# CD103⁻CD8⁺ T cells promote neurotoxic inflammation in Alzheimer's disease via granzyme K−PAR-1 signaling

Eleonora Terrabuio [1,2] ✉, Enrica Caterina Pietronigro [1], Alessandro Bani[1], Vittorina Della Bianca[1], Carlo Laudanna [1,2], Barbara Rossi [1], Giulia Finotti[3], Bruno Santos-Lima[1], Elena Zenaro [1], Ermanna Turano[4], Gabriele Tosadori[1], Matteo Calgaro [5], Nicola Vitulo [5], Monica Castellucci[3], Daniela Cecconi [2,5], Jessica Brandi [5], Nikolaos Vareltzakis [1], Fabiana Mainieri[1], Antonella Calore [1], Gabriele Angelini[1], Bruno Bonetti[6] & Gabriela Constantin [1,2] ✉

Immune mechanisms contribute to the neuropathology of Alzheimer's disease (AD) but the role of adaptive immune cells is unclear. Here we show that the brain CD8⁺ T cell compartment is dysregulated in AD patients and in the 3xTg-AD mouse model, accumulating activated CD103⁻ tissue-resident memory T cells that produce large amounts of granzyme K (GrK). These CD103⁻CD8⁺ T cells originate from the circulation and migrate into the brain using LFA-1 integrin. Ablation of brain CD103⁻CD8⁺ T cells in 3xTg-AD mice ameliorates cognitive decline and reduces neuropathology. GrK induces neuronal dysfunction and tau hyperphosphorylation in human and mouse cells via protease-activated receptor-1 (PAR-1), which is expressed at higher levels in the AD brain, revealing a key immune-mediated neurotoxic axis. We conclude that communication between CD8⁺ T cells and the nervous system is altered in AD, paving the way for therapies targeting T cell-dependent neurotoxic inflammation.

The dysregulation of innate and adaptive immunity is a driving force in the development of Alzheimer's disease (AD), the most common form of dementia, affecting more than 32 million people worldwide[1]. Classical AD neuropathology is characterized by β-amyloid (Aβ) deposition, tau hyperphosphorylation, and the loss of neurons and synapses, but more recent evidence shows that chronic inflammation promoted by local and peripheral immune cells is also a hallmark of AD[2,3].

CD8⁺ T cells are one of the most intriguing components of the AD immunological landscape. During immune responses, these cytotoxic cells produce classical granzymes (such as GrA and GrB), perforin, and a plethora of cytokines[4]. Most CD8⁺ T cells in healthy non-lymphoid organs have a tissue-resident memory (Trm) phenotype, fulfill a local protective role, and can rapidly mount immune responses[5]. However, CD8⁺ T cells also infiltrate the brains of AD patients and equivalent animal models[6–13]. Aging is the main risk factor for AD and recent studies have shown that CD8⁺ T cells also accumulate as the brain ages, promoting axonal degeneration by releasing GrB[14]. But although CD8⁺ T cells are found in close proximity to neuronal structures in AD, it is unclear whether this communication between adaptive immune cells and neural cells promotes disease development[9]. Furthermore, recent studies suggest that the meningeal compartment, which is highly enriched in immune cell populations, may also contribute to AD, but the role of meningeal CD8⁺ T cells in disease development is unclear[15].

[1]Department of Medicine, University of Verona, Strada le Grazie 8, Verona, Italy. [2]The Center for Biomedical Computing (CBMC), University of Verona, Verona, Italy. [3]Centro Piattaforme Tecnologiche (CPT), University of Verona, Verona, Italy. [4]Department of Neuroscience, Biomedicine and Movement Sciences, University of Verona, Verona, Italy. [5]Department of Biotechnology, University of Verona, Strada Le Grazie 15, Verona, Italy. [6]Neurology Unit A, Azienda Ospedaliera Universitaria Integrata of Verona, P. le Stefani, Verona, Italy. ✉e-mail: eleonora.terrabuio@univr.it; gabriela.constantin@univr.it

In most previous studies of AD brains, CD8[+] T cells have been considered as a single homogeneous population[6–8,10]. However, there is clear evidence in other tissues to support the presence of CD8[+] T cell subsets with district roles during homeostasis and immune responses[4]. The population of CD8[+]CD45RA[+]CCR7[-] T effector memory (T[EMRA]) cells expands in the peripheral blood of individuals with mild cognitive impairment (MCI) and AD, and in the cerebrospinal fluid (CSF) following infection with Epstein–Barr virus (EBV), suggesting particular CD8[+] T cell phenotypes may also play a role in AD[9]. Furthermore, CD69[+]CD103[+] and CD69[+]CD103[-]CD8[+] Trm cells have been observed in the human brain[16], but the findings involved a heterogeneous group of subjects with multiple sclerosis (MS), various types of dementia and bipolar disorders, as well as controls with no brain disease. AD patients formed a minority in this mixed group, so it was not possible to determine how CD8[+] Trm cell populations are represented in the AD brain.

The presence of CD8[+] T cell subsets has also been proposed in transgenic animal models of AD. However, these studies involved mice with separate Aβ or tau pathology and showed that CD8[+] T cells may have protective or deleterious effects depending on the disease stage and type of pathology[7,10–13,17]. In APP/PS1 mice with late-stage Aβ pathology, the abundance of CD103[+]CD8[+] T cells increased, and their transcriptomic profile was similar to wild-type (WT) mice, potentially explaining why the depletion of CD8[+] T cells does not affect amyloid pathology[10,11]. The Trm marker CD103 (αE integrin) is also more abundant in the 5xFAD mouse model, which develops an early and aggressive amyloid pathology, together with higher levels of CXCR6 and PD-1 compared to WT controls, and this CD8[+] T cell subset has a protective anti-Aβ role suppressing the activation of microglia[13]. Although CD8[+] T cells generally seem to be deleterious in a model of pure tauopathy[17], the role of specific subsets is less clear. P301S mice expressing APOE4 accumulate activated CD11c[+], KLRE1[+] and ISG15[+] CD8[+] T cells while the pool of TOX[+]PDCD1[+] exhausted CD8[+] T cells is depleted. However, P301S mice lacking APOE show no increase in the abundance of brain CD8[+] T cells despite tauopathy, suggesting additional signals are needed for CD8[+] T cell responses in AD[12].

Aβ and tau pathologies have a synergistic effect in AD and diagnosis requires the presence of both hallmarks[18–20]. Here we performed single-cell RNA sequencing (scRNAseq) in the 3xTg-AD mouse model, which develops both amyloid and tau pathologies, to investigate how brain and meningeal CD8[+] T cell subsets may shape the neuropathology of AD. We found that CD8[+] Trm cells are strongly dysregulated in 3xTg-AD mice, with the number of activated CD103[-] cells increasing as the CD103 expression level declines in the CD8[+] T cell population. We showed that the brain is the main site of CD8[+] T cell dysregulation and found that CD103[-]CD8[+] T cells originate from the circulation and invade the brain using a mechanism dependent on integrin LFA-1. We also observed higher levels of granzyme K (GrK) in the CD103[-]CD8[+] T cells of 3xTg-AD mice and patients with AD compared to controls and discovered a role for GrK in the induction of neuronal dysfunction through the activation of protease-activated receptor 1 (PAR-1). Finally, we demonstrated that GrK–PAR-1 interaction induces tau hyperphosphorylation, revealing a critical immune-mediated neurotoxic axis. Together, our data show that dysfunctional communication between the immune system and central nervous system (CNS) mediated by CD103[-]CD8[+] T cells and GrK–PAR-1 signaling contributes to the development of AD, identifying key molecular mechanisms that can be targeted to prevent immune-mediated neurotoxic inflammation.

## Results

### CD103[-]CD8[+] Trm cells accumulate in the brains of 3xTg-AD mice

Previous studies suggested that CD8[+] T cells play a detrimental role in AD, but the subsets that drive the disease and the underlying mechanisms that contribute to disease development are still unclear[6–12]. We therefore compared the phenotypes and functions of CD8[+] T cells in WT and 3xTg-AD transgenic mice, the latter developing amyloid and tau pathologies representing neuropathological characterics of human AD patients. Single-cell RNA sequencing (scRNAseq) of CD45[HIGH] leukocytes isolated from the meninges and brains of 3xTg-AD (n = 8) mice as well as sex and age-matched WT controls (n = 8) at the onset of cognitive deficit[21,22] (Fig. 1a) revealed 13 cell types, including CD8[+] and CD4[+] T cells, neutrophils, B cells, natural killer (NK) cells, innate lymphoid cells (ILCs), macrophages, and microglia (Fig. 1b; Supp. Fig. 1a-c). Clustering analysis applied to the whole CD8[+] T cell population (3,098 cells representing 19.06% of all CD45[HIGH] leukocytes; Supp. Fig. 1c) revealed the presence of five subsets (Fig. 1c), each characterized by the strong expression of known marker genes (Fig. 1d-f; Supp. Data 1): (i) T central memory (Tcm) cells (4.52%) expressing *Sell*, *Ccr7*, *Nsg2*, *Dapl1* and *Cmah*[23,24]; (ii) T effector cells (11.43%) expressing *Gzma*, *S1pr5*, *Cx3cr1*, *Zeb2*, *Klrg1* and *Klf2*[24,25]; (iii) CD103[-] Trm cells (64.59%) expressing *Gzmk*, *Cxcr6*, *Cxcr3*, *Ltb*, *Xcl1*, *Tnf* and *Eomes*[26,27]; (iv) proliferating Trm (Trm[PROL]) cells (4.29%) expressing *Chek1*, *Stmn1*, *Cdk1*, *Mki67*, *Mcm2*, *Mcm5*, *Mcm7*, *Tfdp1* and *Pola2*[28,29] and (v) CD103[+] Trm cells (15.24%) expressing *Gstp3*, *Foxo1*, *Il10*, *Il17a* and *Il2ra*, as well as *Itgae*, encoding CD103, which is related to the tissue residence phenotype[30–32]. *Cd69*, previously associated with the tissue residence phenotype but also with the activation of T cells[5], was expressed by most of the CD8[+] T cells in both 3xTg-AD and WT mice (Supp. Fig. 1d). Flow cytometry experiments confirmed this phenotypic characterization of CD8[+] T cells showing that: (i) Tcm highly expressed CD62L, encoded by *Sell* gene, and were negative for CD69; (ii) T effector highly expressed CX3CR1, granzyme A (GrA) and granzyme B (GrB), and were negative for CD69; (iii) CD103[-] Trm cells were positive for CD69[+] and expressed high levels of Eomes; and (iv) CD103[+] Trm cells were positive for CD69[+] and expressed low levels of Eomes (Fig. 2a, b; Supp. Fig. 5f–h). Very few Trm[PROL] cells were found in the scRNAseq dataset and these cells were not characterized by flow cytometry.

To better understand differences in the CD8[+] population between 3xTg-AD and WT mice, we separately analyzed the phenotypic changes in the brain and meningeal compartments. Notably, scRNAseq analysis revealed an almost three-fold reduction in the abundance of CD103[+]CD8[+] Trm cells in the brains of 3xTg-AD mice (WT = 27.44%, 3xTg-AD = 9.8%) with a parallel increase in the abundance of CD103[-]CD8[+] Trm cells (WT = 45.9%, 3xTg-AD = 73.69%) (Fig. 2c). Notably, the probability to observe CD103[-]CD8[+] Trm cells was significantly higher (OR = 3.30; P-value = 0) in the brain of 3xTg-AD mice compared to those of WT controls (Data source Fig. 2). In contrast, the probability to observe CD103[+]CD8[+] Trm cells was significantly lower (OR = 0.29; P-value = 0.0038) in the brain of 3xTg-AD mice compared to those of WT controls (Data source Fig. 2). However, these changes were not evident in the meninges, where there was no significant accumulation of CD103[-] cells (OR = 0.72; P-value = 0.09) and the proportion of CD103[+] cells (WT = 11.97%, 3xTg- AD = 19.01%; OR = 1.72; P-value = 0) increased only slightly (Supp. Fig. 1e; Data source Supp. Fig.1). These data suggest that the major alterations in the CD8[+] Trm cell compartment occur in the brain, as corroborated by data showing the downregulation of several transcription factor (TF) genes related to the CD103[+] tissue residence phenotype (*Fabp5*, *Stat1*, *Stat3*, *Stat4*, *Stat5a*, *Stat5b*, *Smad4*, *Smad6*, *Smad7* and *Smad9*) and the suppression of genes encoding key cytokines involved in the differentiation and maintenance of CD103[+] Trm cells (*Il15*, *Il7*, *Tgfb3*, *Tgfb2*, *Tgfb1*, *Il21*, *Tnf* and *Il33*; Fig. 2d, f). Moreover, the expression level of *Itgae* was reduced, whereas the expression of *Eomes* and *S1pr1*, which inhibits the CD103[+]CD8[+] T cell phenotype[5,31], was upregulated in CD8[+] T cells from the brains of 3xTg-AD mice compared to WT controls (Fig. 2e). These data highlight the dysregulation of brain CD8[+] T cells and the depletion of the CD103[+] Trm population, presumably affecting CNS immunity and contributing to disease development.

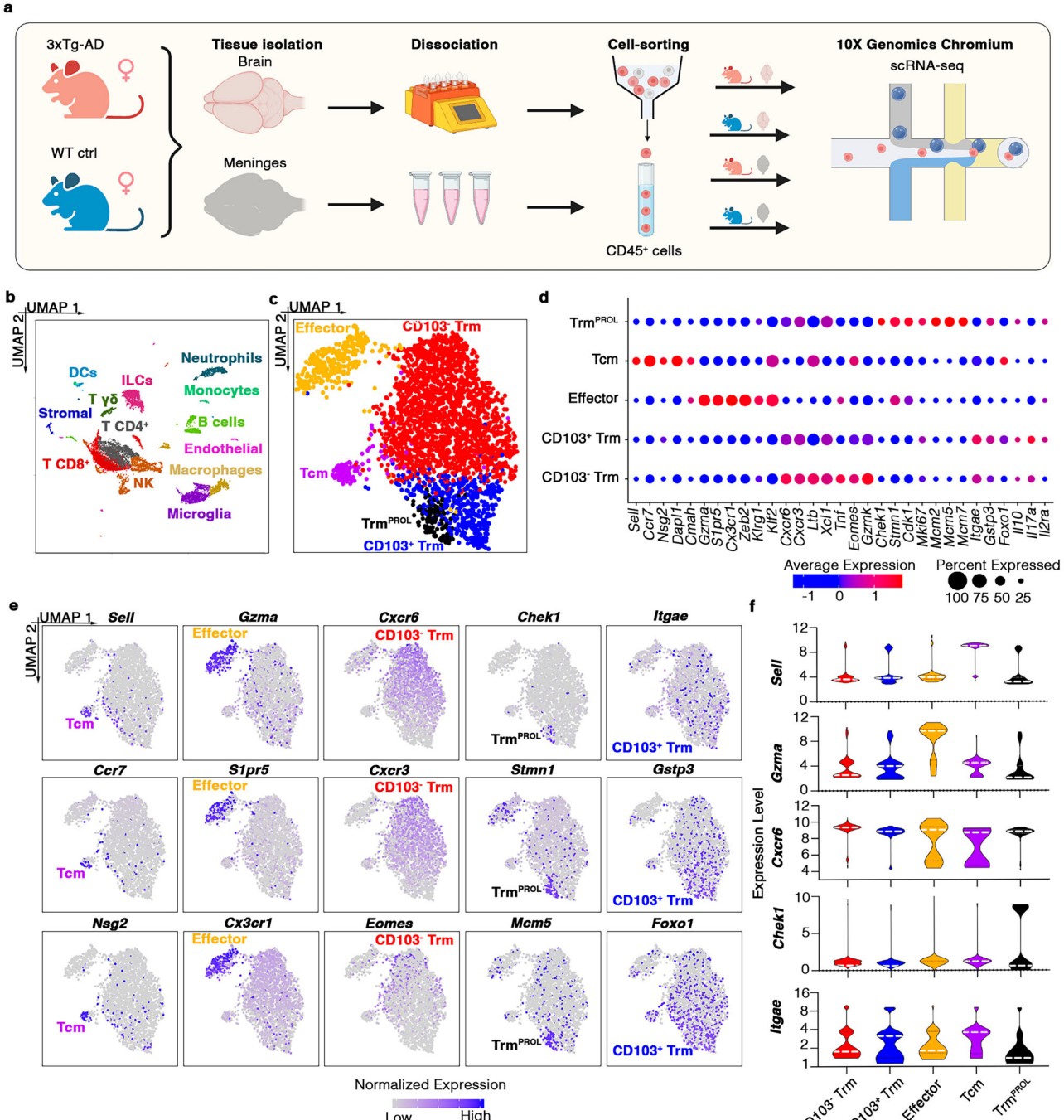

**Fig. 1 | Characterization of CD8 T cell phenotype in 3xTg-AD and WT mice.**
**a** Graphical overview of the scRNAseq experimental design created in BioRender.
Terrabuio, E. (2025) https://BioRender.com/f3lmkb2. **b** UMAP plot showing
CD45^HIGH leukocytes detected in the brains and meninges of wild-type (WT; $n=8$)
and 3xTg-AD ($n=8$) mice. CD8⁺ T cells ($n=3{,}098$) are shown in red. **c** UMAP plots
showing the five subsets of CD8⁺ T cells detected in the brains and meninges of WT

($n=8$) and 3xTg-AD ($n=8$) mice. **d** Bubble plot reporting the phenotypic marker
genes for each CD8⁺ T cell subset. Transcript levels are color-coded. **e** Normalized
expression of known marker genes on UMAP plot. Transcript levels are color-
coded. **f** Violin plots showing the expression of marker genes for each CD8⁺ T cell
subset. White dashed line indicates the median expression level.

The results above were supported by unbiased cell fate trajectory
analysis showing that CD8⁺ T cell differentiation followed a tightly
organized trajectory along with the pseudo-time progression (Fig. 3a),
starting from a common root and ending with two differentiation
states distinguished by *Itgae* (Arm A) and *Eomes* (Arm B) expression
(Fig. 3b). Only slight differences were observed in the meninges
whereas there were clear differences in the brain (Fig. 3c, Supp.
Fig. 1f–l), where Arm A of the trajectory plot was populated by a lower
proportion of CD103⁺CD8⁺ Trm cells in 3xTg-AD mice compared to WT

controls (WT = 26.1%, 3xTg-AD = 14.9%) accompanied by a higher per-
centage of CD103⁻CD8⁺ Trm cells in Arm B (WT = 54.5%, 3xTg-AD =
75.8%; Fig. 3c; Supp. Fig. 1h-l). This supports the hypothesis that the
brain is a fundamental point of CD8⁺ T cell dysregulation in 3xTg-
AD mice.

Flow cytometry showed that most brain CD8⁺ T cells were CD69⁺
(Supp. Fig. 1m) and clearly validated the presence of immune dysre-
gulation in the brain CD8⁺ Trm cell compartment. We observed a dra-
matic increase in both the proportion and absolute numbers of

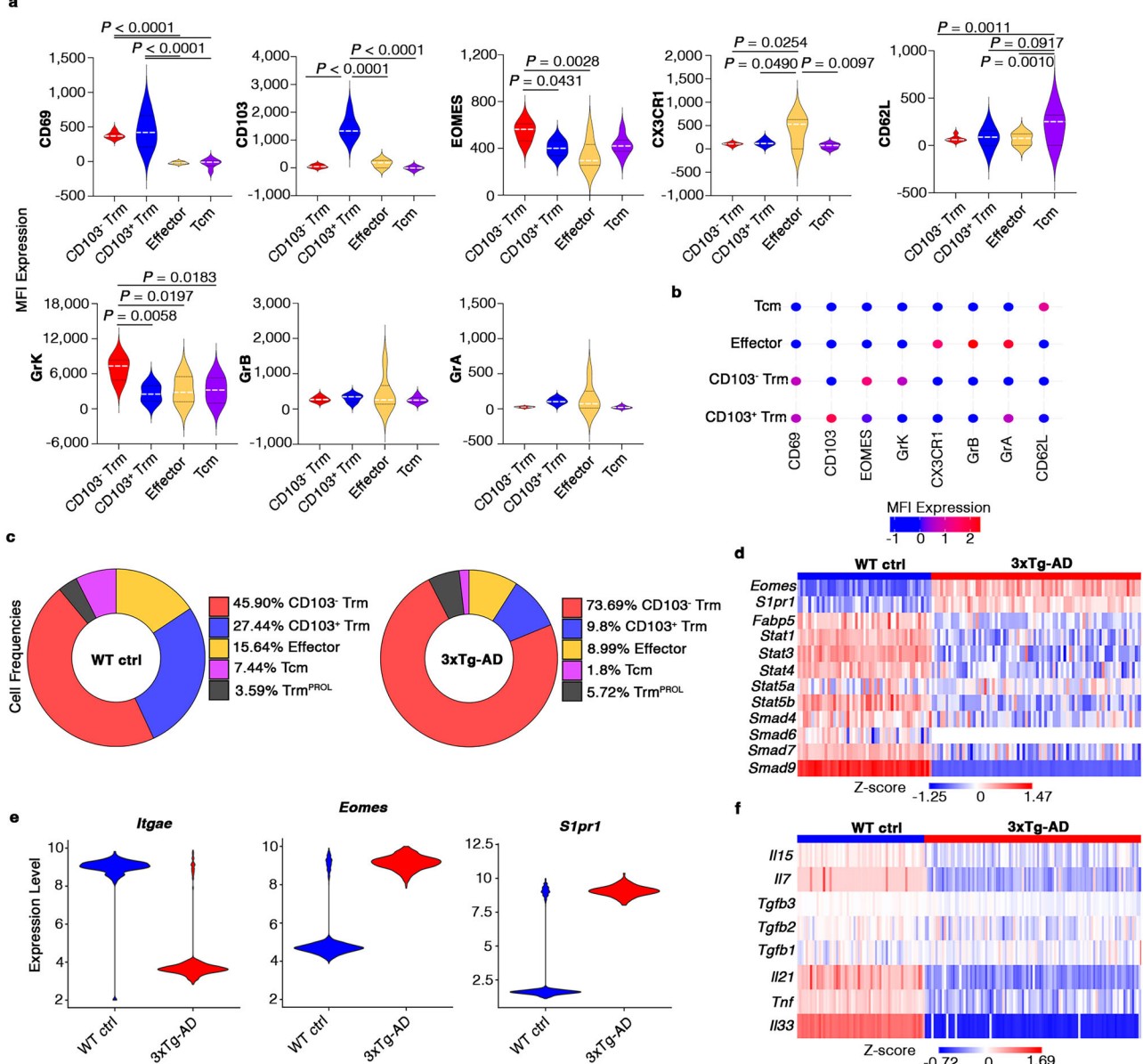

**Fig. 2 | The CD8+ Trm compartment is altered in the brains of 3xTg-AD mice.**
**a**, **b** Violin plots (g) and bubble plot (h) showing the expression of phenotypic markers detected by flow cytometry in the brains and meninges of WT (*n* = 3 pools of two organs each) and 3xTg-AD (*n* = 3 pools of two organs each) mice. White dashed line indicates the median expression level in violin plots (g). MFI expressions in the bubble plot (h) are scaled and color-coded. *P*-values are based on two-way ANOVA - multiple comparisons. Source data are provided as a Source Data file.

**c** Donut plots indicating the distribution of CD8+ T cell subsets in the brains of WT (left) and 3xTg-AD (right) mice **d** Heat map showing the expression of residency genes in the brains and meninges of WT and 3xTg-AD mice. Transcript levels are color-coded. **e** Violin plots showing gene expression in the brains of WT (*n* = 8) and 3xTg-AD (*n* = 8) mice. **f** Heat map showing the expression in CD45^HIGH leukocytes of genes in brains of WT (*n* = 8) and 3xTg- AD (*n* = 8) mice. Transcript levels are color-coded.

CD69+CD103− Trm cells in the brains of 6-month-old 3xTg-AD mice compared to sex/age-matched WT controls (WT proportion = 48.025%, 3xTg-AD = 88.08%; WT number = 6,224.72, 3xTg-AD = 12,760.5), with CD69+CD103+ Trm cells showing the opposite profile (Fig. 3d, e). However, we detected no significant differences between genotypes in the numbers of CD103−CD8+ Trm cells populating the meninges (WT number = 9,018.66, 3xTg-AD = 8,895.18), highlighting the brain as a key site for pathological changes in 3xTg-AD mice (Supp. Fig. 1n, o).

**Brain CD103−CD8+ T cells show an activated phenotype strongly associated with AD**
Gene set enrichment analysis (GSEA) using the Kyoto Encyclopedia of Genes and Genomes (KEGG) database reported "pathway of neurodegeneration – multiple diseases", "Alzheimer's disease" (AD), and

"Huntington disease" (HD) as the three most enriched terms in the brains of 3xTg-AD mice (Supp. Fig. 1p), whereas these pathways were not enriched in the meninges (Supp. Fig. 1q). Importantly, whereas CD103+CD8+ Trm cells were negatively associated with the AD pathway suggesting a protective role, CD103−CD8+ Trm cells infiltrating the brains of 3xTg-AD mice showed a strong positive association with this pathway (Supp. Data 1), further suggesting these cells may be specifically involved in AD (Fig. 3f). Moreover, GSEA applied to the biological process Gene Ontology (BP-GO) database revealed an activated phenotype for CD103−CD8+ Trm cells populating the brains of 3xTg-AD mice compared to WT controls. Indeed, the five most enriched biological processes included "*Immune system process*" (GO:0002376) and "*Regulation of immune system process*" (GO:0002682), indicating pathways involved in the development of

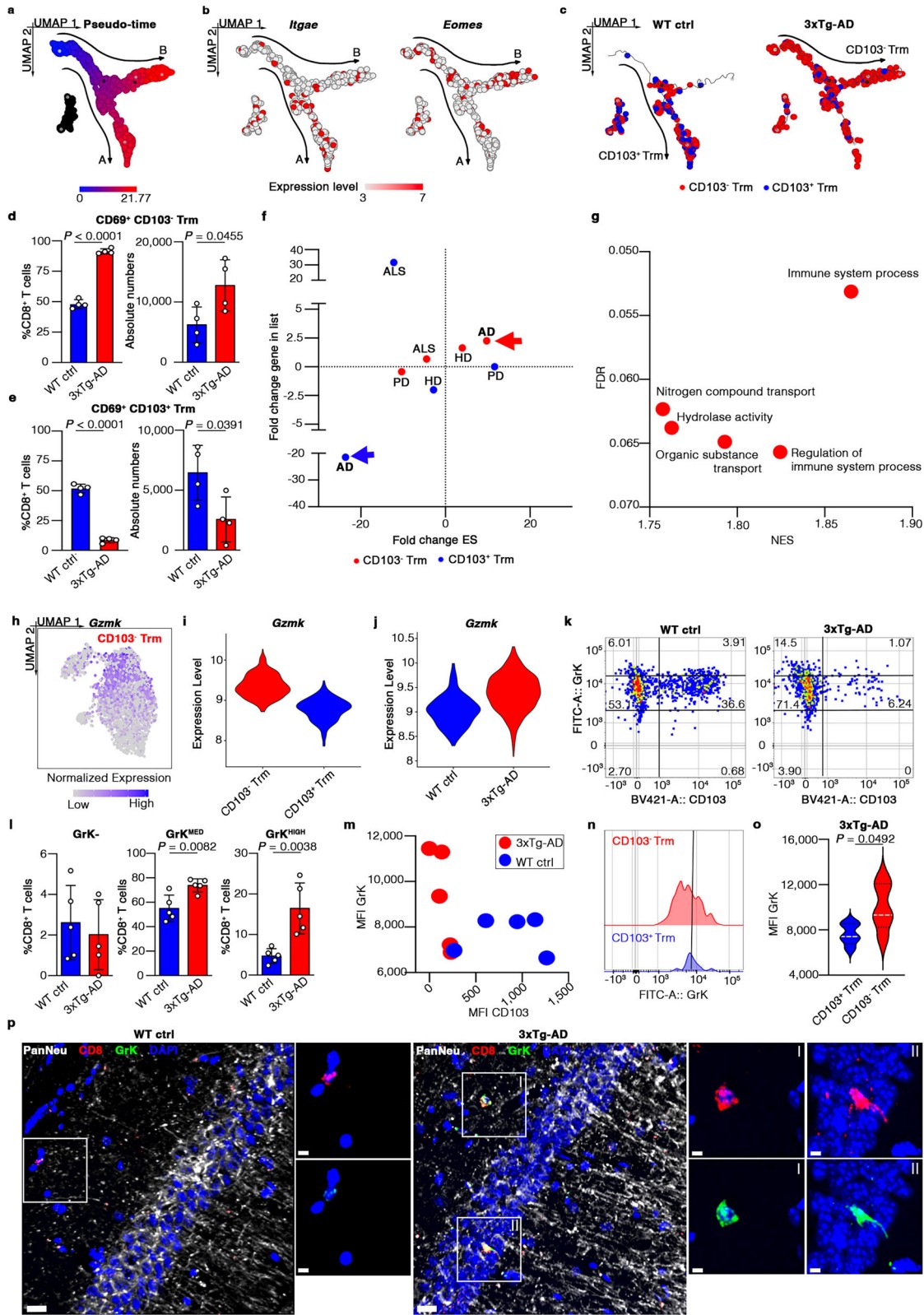

immune responses, "*Nitrogen compound transport*" (GO:0071705), which is related to T cell activation[33], and "*Hydrolase activity*" (GO:0016787), which mediates the secretion of cytotoxic granules, suggesting a potential pro-inflammatory and cytotoxic role for CD103⁻CD8⁺ T cells in AD[34] (Fig. 3g). Collectively, our data show a profound dysregulation of the CD8⁺ Trm cell compartment in the brains of 3xTg-AD mice, with a loss of CD103⁺ cells and the

accumulation of activated CD103⁻ cells potentially contributing to neurotoxicity and AD development.

## CD103⁻CD8⁺ Trm cells produce granzyme K in the brains of 3xTg-AD mice

We next sought to identify the pathological mechanisms mediated by activated CD103⁻ cells under AD-like conditions and found that *Gzmk*,

**Fig. 3 | GrK expression is upregulated in brain CD103⁻CD8⁺ Trm cells of 3xTg-AD mice. a** Pseudo-time ordered trajectory plot of CD8⁺ T cells in the brains and meninges of WT and 3xTg- AD mice. **b** Expression of genes in the trajectory plots. Transcript levels are color-coded. **c** Trajectory plot indicating the distribution of brain CD103⁻ and CD103⁺CD8⁺ Trm cells in WT and 3xTg-AD mice. **d, e** Abundances of brain CD103⁻ (d) and CD103⁺ (e) CD8⁺ Trm cells in WT ($n = 4$) and 3xTg-AD ($n = 4$) mice. Data are means ± SD. *P*-values based on two-tailed Student's t-test. **f** Scatter plot showing KEGG pathway enrichment analysis for CD103⁻ and CD103⁺ CD8⁺ Trm cells in the brains of 3xTg-AD mice (PD = "*Parkinson disease*", ALS = "*Amyotrophic lateral sclerosis*"). **g** Bubble plot showing BP-GO GSEA analysis for brain CD103⁻CD8⁺ Trm cells in 3xTg-AD mice. **h** Normalized gene expression on UMAP plot of brain and meningeal CD8⁺ T cells. Transcript levels are color-coded. **i** Violin plots showing gene expression in CD103⁻ and CD103⁺CD8⁺ Trm cells in the brain and meninges.

**j** Violin plot showing gene expression in CD8⁺ T cells in the brains of WT and 3xTg-AD mice. **k** Representative flow cytometry plot showing GrK expression in brain CD8⁺ T cells in WT and 3xTg-AD mice. **l** Bar plots showing the percentage of GrK⁻, GrK^MED, and GrK^HIGH CD103⁻ CD8⁺ Trm cells in the brains of WT ($n = 5$) and 3xTg-AD ($n = 5$) mice. Data are means ± SD. *P*-values based on two-tailed Student's t-test. **m** Scatter plot showing MFIs of CD103 and GrK in CD8⁺ T cells in the brains of WT ($n = 5$) and 3xTg-AD ($n = 5$) mice. **n, o** Representative histograms (n) and violin plots (o) showing GrK MFI for brain CD103⁻ (red) and CD103⁺ (blue) CD8⁺ Trm cells in 3xTg-AD mice. White dashed lines indicate median expressions. *P*-values based on two-tailed Student's t-test. **p** CD8⁺ T cells (squares) in the brains of WT and 3xTg-AD mice detected by immunofluorescence microscopy. Scale bar = 5 μm or 2 μm for zoomed images.

encoding GrK, was one of the most strongly expressed marker genes in the global CD103⁻CD8⁺ Trm cell population (brain and meninges; Fig. 3h, i; Supp. Fig. 2a; Supp. Data 1). Interestingly, the genes encoding other granzymes (*Gzma* and *Gzmb*) and perforin-1 (*Prf1*) were not found in the list of marker genes in CD103⁻CD8⁺ Trm cells compared to other subsets (Supp. Data 1). Flow cytometry experiments validated these data confirming the low expression of GrA and GrB but high expression of GrK in the CD103⁻ CD8⁺ Trm population (Fig. 2a, b), suggesting GrK may have a selective functional role in the CD103⁻ Trm cell subpopulation in AD.

Importantly, *Gzmk* expression was upregulated in the brains but not in the meninges of 3xTg-AD mice compared to WT controls (Fig. 3j; Supp. Fig. 2b). Flow cytometry confirmed the scRNAseq data, showing a dramatic increase in the percentage of GrK-producing CD103⁻CD8⁺ Trm cells (both GrK^MED and GrK^HIGH) infiltrating the brain, but not the meninges, of 3xTg-AD mice compared to WT controls, whereas no differences were observed for brain-infiltrating CD103⁻GrK⁻ CD8⁺ T cells (Fig. 3k–m; Supp. Fig. 2c). Importantly, we also found that the strong increase in intracellular GrK expression was selective for the CD103⁻CD8⁺ Trm cell population (86% of which produced GrK) when compared to the CD103⁺ population (Fig. 3n, o), suggesting the production of GrK is a key pathological mechanism of CD103⁻CD8⁺ Trm cells in the brains of 3xTg-AD mice.

## GrK⁺CD103⁻CD8⁺ Trm cells induce neuronal dysfunction via the GrK−PAR-1 axis

GrK is associated with pro-inflammatory and cytotoxic activities[35], suggesting that GrK produced by CD103⁻CD8⁺ T cells may induce neuronal dysfunction in AD. Immunofluorescence staining revealed the presence of GrK⁺CD8⁺ T cells in close proximity to hippocampal neurons in the brains of 3xTg- AD mice, whereas CD8⁺ T cells with lower GrK expression were predominantly detected in WT brains (Fig. 3p; Supp. Fig. 2j). These results indicate that GrK⁺CD103⁻CD8⁺ Trm cells may contribute to neuronal alterations in 3xTg-AD mice.

To confirm that GrK⁺CD103⁻CD8⁺ Trm cells directly induce neuronal dysfunction in AD, we used wide-field high-resolution microscopy for the live imaging of primary hippocampal neurons isolated from 3xTg-AD mice co-cultured with CD103⁻ or CD103⁺ CD8⁺ Trm cells obtained from the livers of 3xTg-AD mice, an abundant source of CD8⁺ Trm cells[36] (Fig. 4a). Immunofluorescence staining confirmed the high density of GrK granules inside CD103⁻CD8⁺ Trm cells compared to CD103⁺ counterparts (Fig. 4b). Time-lapse fluorescence microscopy revealed that primary neurons in contact with GrK⁺CD103⁻CD8⁺ Trm cells, but not those in contact with CD103⁺CD8⁺ Trm cells, showed significantly higher cytoplasmic calcium (Ca²⁺) levels compared to the negative control, clearly indicating that GrK⁺CD103⁻CD8⁺ Trm cells induce intracellular Ca²⁺ dysregulation (Fig. 4c), which is associated with neuronal alterations[37]. Importantly, purified active GrK directly induced intracellular Ca²⁺ release in a dose-dependent manner, unequivocally demonstrating the detrimental role of this enzyme in the context of neuronal dysfunction (Fig. 4d).

GrK can bind the thrombin receptor PAR-1[38], which is implicated in synaptic plasticity and memory formation in healthy mice, but also plays a negative role in some brain pathologies[39–41]. Importantly, immunofluorescence staining revealed a significant increase in both the number and area of PAR-1⁺ neurons in the hippocampus of 3xTg-AD mice compared to WT controls, suggesting that signaling via PAR-1, a G-protein coupled receptor (GPCR), is dysregulated in AD (Supp. Fig. 2g, h). Importantly, wide-field time-lapse microscopy showed that the PAR-1 inhibitor SCH79797 strongly reduced intra-neuronal Ca²⁺ release in the presence of GrK⁺CD103⁻CD8⁺ Trm cells (Fig. 4e). Moreover, our data clearly showed the strong, dose-dependent inhibition of intracellular Ca²⁺ release in neurons cultured with 150 nM purified active GrK in the presence of SCH79797 (Fig. 4f, g; Supplementary Movies 1-3). Immunofluorescence staining confirmed the presence of GrK⁺CD103⁻CD8⁺ Trm cells in close contact with PAR-1⁺ neurons in in vitro cultures (Fig. 4h), and of GrK⁺ CD8⁺ T cells in the hippocampus of 3xTg-AD mice (Supp. Fig. 2i), further supporting a role for GrK−PAR-1 binding in the neuronal dysfunction underlying AD. These data demonstrate that the GrK−PAR-1 axis is a key immune-mediated pathological mechanism promoting neuronal dysfunction in AD.

## Brain CD103⁻CD8⁺ Trm cells originate from the circulation and are detrimental in 3xTg-AD mice

The intraperitoneal (ip) treatment of mice with an anti-CD8 antibody depletes circulating CD8⁺ T cells leaving the brain Trm compartment unaltered[42,43]. To determine the origin of CD103⁻ T cells, we therefore used this method to deplete circulating CD8⁺ T cells in 3xTg-AD mice and sex-matched WT controls (Supp. Fig. 3a-h). The anti-CD8 treatment significantly reduced the abundance of brain CD103⁻CD8⁺ Trm cells compared to mice treated with an isotype control antibody, reaching the levels of the control groups (Fig. 5a), suggesting this cell subset is replenished from the blood. However, we detected no differences in the abundance of CD103⁺CD8⁺ Trm cells after CD8⁺ T cell depletion, in 3xTg-AD mice and WT controls (Fig. 5a), in line with previous data showing these cells are spared by systemic depletion[43]. Importantly, the ablation of CD103⁻CD8⁺ Trm cells in the brains of 3xTg-AD mice was paralleled by an amelioration of cognitive functions in behavioral tests, confirming that CD103⁻CD8⁺ Trm cells contribute to memory decline in 3xTg-AD mice. Particularly, the Morris water maze (MWM) test showed improved hippocampal-dependent learning after CD8⁺ T cell depletion in 3xTg-AD mice compared to controls, as indicated by the learning curve recorded during the training days of the test (Fig. 5b). In addition, the probe test showed that 3xTg-AD mice devoid of circulating CD8⁺ T cells actively searched the platform, maintaining the memory of the original platform position and achieving better escape performance, as also shown by the greater path efficiency and significant reduction in body rotations (Fig. 5c−e). The contextual fear conditioning (CFC) associative learning task revealed an amelioration of associative memory following the depletion of circulating CD8⁺ T cells in 3xTg-AD mice, as shown by the

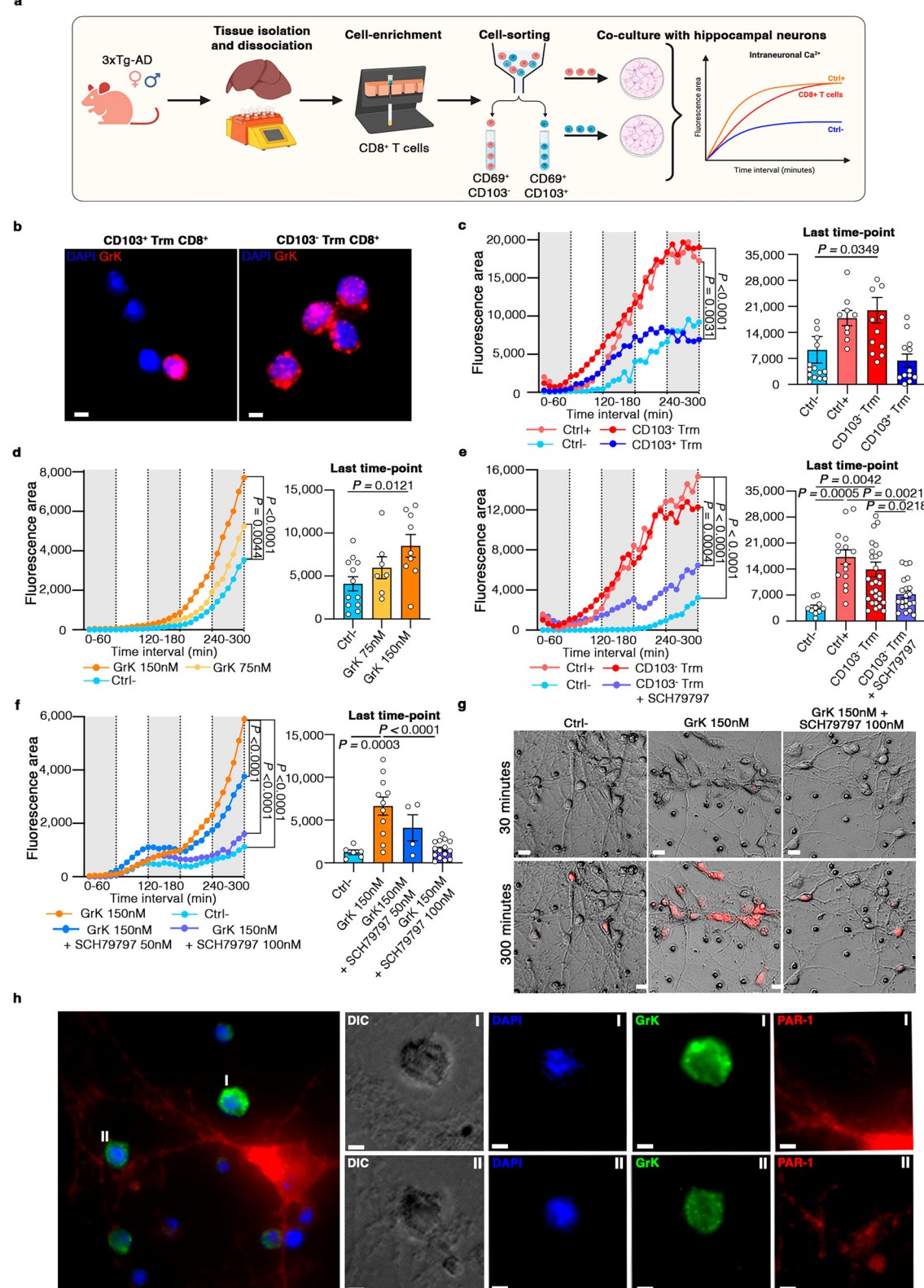

significant increase in the percentage of freezing responses compared to control animals treated with an isotype control (Fig. 5f).

To confirm whether the amelioration of cognitive deficits by anti-CD8a antibody treatment correlates with the loss of neuropathological hallmarks of AD, we stained coronal murine brain sections with anti-Aβ (6E10), anti-phospho-tau (AT180), and anti-total-tau (HT7) antibodies. We observed a significant decrease in both the Aβ load and levels of tau

hyperphosphorylation in the hippocampus of 3xTg-AD mice following the depletion of circulating CD8+ T cells compared to animals treated with an isotype control (Fig. 5g, h), whereas the levels of total tau were unchanged (Fig. 5i). These data were confirmed by ELISA experiments, showing significantly less Aβ1-40 and Aβ1-42 deposition, and tau hyperphosphorylation (pT231, AT180) in the soluble and insoluble fractions of brain homogenates from 3xTg-AD mice depleted of

**Fig. 4 | GrK⁺CD103⁻CD8⁺ Trm cells induce neuronal alterations by engaging PAR-1. a** Graphical overview of the experimental design. Created in BioRender. Terrabuio, E. (2025) https://BioRender.com/ir5h5rf. **b** Immunofluorescence staining showing GrK granules inside CD103⁺ and CD103⁻ CD8⁺ Trm cells. Scale bar = 3 μm. **c** Intracellular Ca²⁺ release in neurons co-cultured with CD103⁺ or CD103⁻ CD8⁺ Trm cells. The last time point is shown in the right panel. Data are means ± SD from four independent experiments. *P*-values based on two-way ANOVA-multiple comparisons. Ctrl⁻ = neurons alone. Ctrl⁺ = ionomycin-stimulated neurons (10 μM). Source data are provided as a Source Data file. **d** Intracellular Ca²⁺ release in neurons treated with purified active GrK or vehicle. The last time point is shown in the right panel. Data are means ± SD from three independent experiments. *P*-values based on two-way ANOVA-multiple comparisons. Source data are provided as a Source Data file. **e** Intracellular Ca²⁺ release in neurons co-cultured for 5 h with CD103⁻CD8⁺ Trm cells in the presence/absence of the PAR-1 inhibitor SCH79797

(100 nM). The last time point (300 min) is shown in the right panel. Data are means ± SD from four independent experiments. *P*-values based on two-way ANOVA-multiple comparisons. Ctrl⁻ = neurons alone. Ctrl⁺ = ionomycin-stimulated neurons (10 μM). Source data are provided as a Source Data file. **f, g** Intracellular Ca²⁺ release in neurons cultured for 5 h with purified active GrK alone (150 nM) or with SCH79797 (50 nM, or 100 nM). Ctrl⁻ = neurons alone. The last time point is shown in the right panel. Data are means ± SD from three independent experiments. *P*-values based on two-way ANOVA-multiple comparisons. Representative images are shown in (g). Red = intracellular Ca²⁺ release. Scale bar = 20 μm. Source data are provided as a Source Data file. **h** Immunofluorescence microscopy showing GrK⁺CD103⁻CD8⁺ Trm cells near the soma (cell I) and dendrites (cell II) of a PAR-1⁺ neuron. Cell morphology was visualized by wide-field imaging using a DIC filter. Scale bar = 5 μm (or 2 μm for zoomed images I, and II).

CD8 + T cells compared to isotype controls (Fig. 5j, k), while no significant differences were observed in the levels of total tau (Fig. 5l). Similarly, we observed significantly lower levels of insoluble oligomeric (A11 antibody) and fibrillar (OC antibody) Aβ forms in the brain of 3xTg-AD mice depleted of CD8 + T cells compared to isotype controls (Fig. 5m, Supp. Fig. 3j, Supp. Data 5-8). Importantly, our immunofluorescence staining showed that the oligomeric and fibrillar forms of Aβ were located at the intraneuronal level in the hippocampus of 9 9-month-old 3xTg-AD mice, and both co-localized with the 6E10 signal (Fig. 5n).

Collectively, these results demonstrate that CD103⁻CD8⁺ Trm cells originate from the circulation, infiltrate the brain, and promote memory decline and neuropathological changes that characterize AD, strongly supporting their role in AD pathogenesis.

## LFA-1 integrin mediates the infiltration of CD103⁻CD8⁺ T cells into the brain

Next, we investigated the molecular mechanisms that allow circulating CD8⁺ T cells to infiltrate the brains of 3xTg-AD mice. Our recent work suggests that α4-integrins are not involved in CD8⁺ T cell extravasation in 3xTg-AD mice[44]. However, our flow cytometry data showed a significant increase in both the percentage and mean fluorescence intensity (MFI) of LFA-1⁺ circulating CD8⁺ T cells in the blood of 3xTg-AD mice compared to age-matched WT controls (Fig. 6a). These results, together with our earlier data showing the upregulation of brain endothelial intracellular adhesion molecule 1 (ICAM-1), the counterligand of LFA-1[22], suggest a role for LFA-1 integrin in the trafficking of CD8⁺ T cells into the brain. To understand the role of LFA-1 in more detail, we crossed 3xTg-AD mice with *Itgal*⁻/⁻ mice lacking functional LFA-1 integrin and observed a significant decrease in the percentage of CD103⁻CD8⁺ Trm cells in the brains of 3xTg-AD/*Itgal*⁻/⁻ mice compared to 3xTg-AD animals (Fig. 6b, c). Neuropathological studies showed a significantly lower Aβ load and significantly less tau hyperphosphorylation (AT180) in the brains of 3xTg-AD/*Itgal*⁻/⁻ mice compared to sex- and age-matched 3xTg-AD controls, but no significant difference in the levels of total tau (Fig. 6d–f). Overall, these data suggest a pivotal role for LFA-1 in the accumulation of CD103⁻CD8⁺ Trm cells in the brain, and further supports the origin of detrimental CD103⁻CD8⁺ Trm cells in the circulation.

## Circulating CD8⁺ T cells express more LFA-1 in AD patients

To understand the translational potential of our data, we also investigated the role of LFA-1 in the trafficking of circulating CD8⁺ T cells in AD patients. We analyzed a published scRNAseq dataset (accession GSE181279) of CD45⁺ leukocytes isolated from the blood of three AD patients and two negative controls (NCs)[45] (Fig. 6g–m). We used known gene signatures to characterize CD8⁺ T cells (*n* = 10,162) in the dataset (Fig. 6i, j), identifying: (i) naïve; (ii) early activated; and (iii) CD8⁺ Tem cells[46] (Fig. 6k, l; Supp. Data 2). Interestingly, these results showed that *ITGAL*, encoding the CD11a subunit of LFA-1 integrin, was upregulated

in the whole population of circulating CD8⁺ T cells in AD patients 1 and 3, compared to controls (Fig. 6m). Particularly, *ITGAL* expression was upregulated in the Tem subset of circulating CD8⁺ T cells in AD patients compared to controls (Fig. 6m) but was not differentially expressed in the early activated or naïve CD8⁺ T cell populations (Fig. 6m). These results agree with our data showing a higher percentage of circulating LFA- 1⁺CD44⁺KLRG1⁺CD8⁺ Tem cells and the stronger expression of LFA-1 on Tem cells in 3xTg-AD mice compared to WT controls (Supp. Fig. 4a), whereas no significant differences were observed in the KLRG1⁻CD44⁺ and KLRG1⁻CD44⁻ subpopulations of circulating CD8⁺ T cells (Supp. Fig. 4b, c). Collectively, these data confirm that circulating activated CD8⁺ T cells, particularly the Tem subset, are more activated and have a greater LFA-1-dependent migration capacity in AD patients, as well as 3xTg-AD mice.

## GrK-producing CD8⁺ T cells accumulate in the CSF and brains of AD patients

We also analyzed a published scRNAseq dataset of CSF immune cells (accession GSE134578)[9] (Fig. 7a, b), revealing two subsets of CD8⁺ T cells (Fig. 7c, d): (i) CD69⁺ and PRDM1⁺ Trm cells[24,47], and (ii) *KLF2*-expressing CD8⁺ T cells (Supp. Data 3). The CD8⁺ Trm population could be divided into two further subpopulations (Fig. 7e; Supp. Data 3): (i) CD103⁻CD8⁺ Trm cells characterized by the expression of *GZMK*, *EOMES,* and *S1PR1*, and (ii) *ITGAE*-expressing CD103⁺ CD8⁺ Trm cells characterized by the expression of *CD7, ITGA1, KLRB1*, and *CCR6*[32]. We observed an increase in the abundance of CD103⁻CD8⁺ Trm cells during disease progression, whereas the number of CD103⁺CD8⁺ Trm cells declined, supporting our data from 3xTg-AD mice (Fig. 7f). Importantly, we also observed a significant increase in *GZMK* expression in CD103⁻CD8⁺ Trm cells in AD patients compared to healthy controls and patients with MCI (Fig. 7 g, h), strongly suggesting that GrK-producing CD103⁻CD8⁺ Trm cells also play a detrimental role in the development of AD in humans.

We used immunofluorescence staining to determine whether GrK-producing CD103⁻CD8⁺ Trm cells also accumulate in the human brain during AD (Supp. Table 1). Importantly, whereas GrK⁻CD103⁺ CD8⁺ T cells were predominantly detected in the hippocampus of controls, we observed the presence of both intraparenchymal and intravascular GrK⁺CD103⁻CD8⁺ T cells near the hippocampal neurons of AD patients (Fig. 8a). These observations match our 3xTg-AD mice, strongly supporting a role for GrK-producing cells in AD. In further agreement with our mouse data, we observed a significant increase in the abundance of intraparenchymal CD103⁻CD8⁺ T cells in the hippocampus of AD patients but a lower percentage of CD103⁺CD8⁺ T cells (Fig. 8b). Although not statistically significant, we observed a slight increase in the percentage of both CD103⁻CD8⁺ and GrK⁺CD103⁻ CD8⁺ T cells at the intravascular level in the hippocampus of AD patients compared to controls (Fig. 8c). Importantly, our quantifications showed a significant increase in the number of intraparenchymal GrK⁺CD103⁻CD8⁺ T cells in the hippocampus of AD patients compared

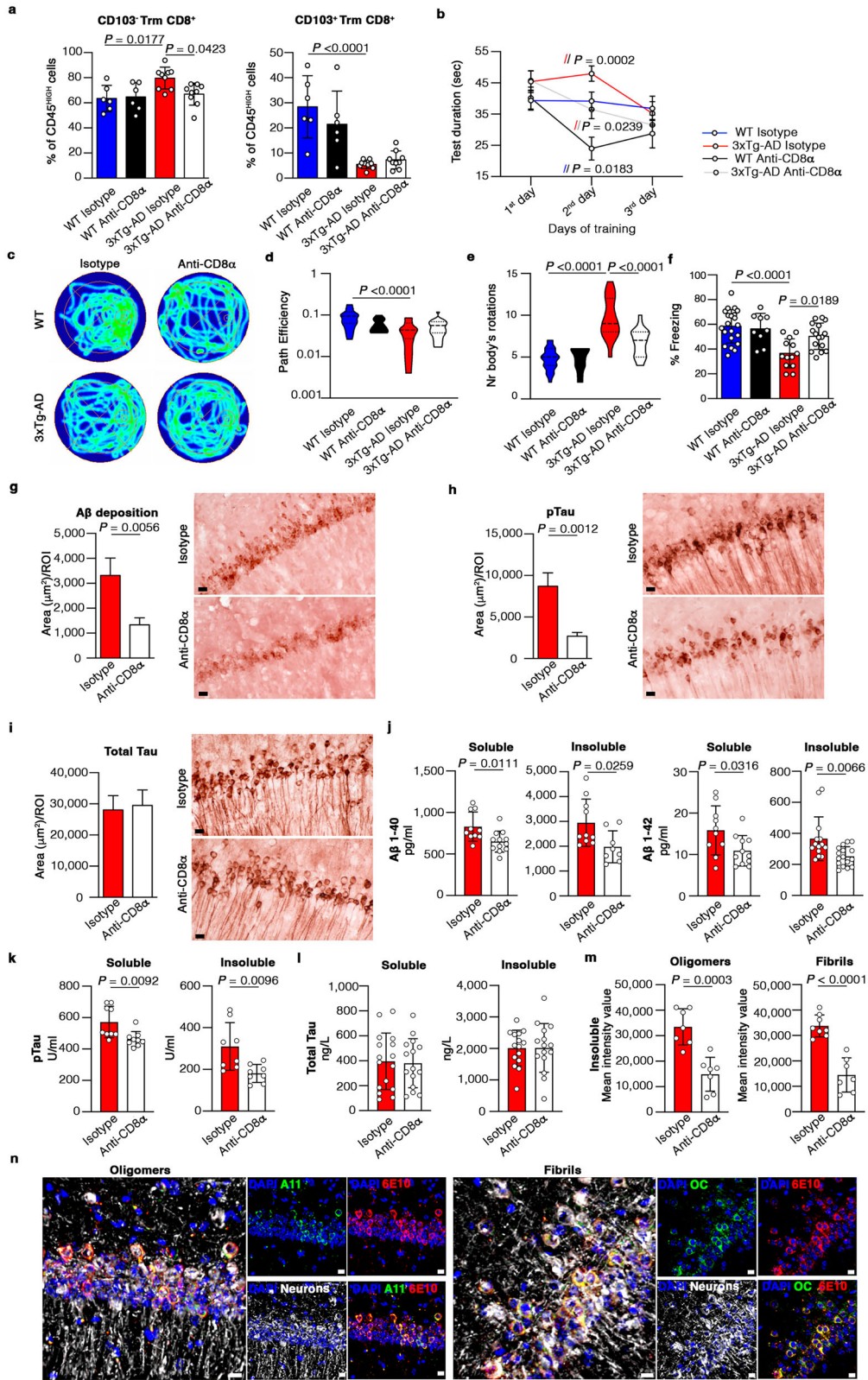

to age-matched controls, suggesting a role for GrK in the neuronal dysfunction associated with AD in humans (Fig. 8b).

## GrK-PAR-1 signaling induces functional alterations and tau hyperphosphorylation in human neuronal cells

Finally, we assessed whether GrK can induce neuronal alterations via PAR-1 in human cells by measuring intracellular $Ca^{2+}$ released by

differentiated SH-SY5Y human neuroblastoma cells treated with purified GrK in the presence or absence of the PAR-1 inhibitor SCH79797 (Fig. 9a–c). We observed a significant increase in the release of intracellular $Ca^{2+}$ by GrK alone, whereas cells cultured with GrK in the presence of SCH79797 showed intracellular $Ca^{2+}$ levels comparable to the negative control (Fig. 9b, c). Next, we showed that recombinant GrK alone significantly increased the hyperphosphorylation of tau on

**Fig. 5 | Detrimental CD103⁻CD8⁺ Trm cells in the brain originate from the circulation and their depletion ameliorates disease in 3xTg-AD mice. a** Flow cytometry showing the percentage of brain CD103⁻ and CD103⁺CD8⁺ Trm cells after anti-CD8 antibody treatment. Anti-CD8 (WT, $n = 6$; 3xTg-AD, $n = 9$), isotype-control (WT, $n = 6$; 3xTg-AD, $n = 9$). Data are means ± SD from two independent experiments and *P*-values are based on one-way ANOVA-multiple comparisons was used. **b** Time spent during training (MWM test). Anti-CD8 (WT, $n = 10$; 3xTg-AD, $n = 21$), isotype-control (WT, $n = 22$; 3xTg-AD, $n = 18$). Data are means ± SD from two-independent experiments (two-way ANOVA-multiple comparisons). **c** Representative tracking of three mice/group (MWM test). **d, e** Violin plots showing path efficiency (d) and number of body rotations (e) (MWM test). Data are from two-independent experiments one-way ANOVA-multiple comparisons. **f** Bar plot showing the percentage of freezing during the CFC test after anti-CD8 treatment. Anti-CD8 (WT, $n = 9$; 3xTg-AD, $n = 16$), isotype-control (WT, $n = 21$; 3xTg-AD,

$n = 13$). Data are means ± SD from two-independent experiments. One-way ANOVA-multiple comparisons was used. **g–i** Immunohistochemical staining of the hippocampus after CD8 T cell depletion showing Aβ-load (g), the levels of hyperphosphorylated (h) and total (i) tau protein. Anti-CD8 ($n = 3$), isotype-control ($n = 3$). Scale bar = 20 μm. Data are means ± SEM. Two-tailed Student's t-test was used. ROI = 624.7 μm×501.22 μm. **j–l** ELISA showing Aβ 1−40 and Aβ 1−42 (j) levels of tau hyperphosphorylation (k) and total tau (l) in soluble and insoluble fractions of brain homogenates after anti-CD8 treatment. Anti-CD8 ($n = 4$), isotype control ($n = 4$). Data from three-independent experiments are means ± SEM (two-tailed Student's t-test). **m** Dot blot showing insoluble oligomeric and fibrillar forms of Aβ in brain homogenates after anti-CD8 treatment. Anti-CD8 ($n = 4$), isotype-control ($n = 4$). Data from three-independent experiments are shown as means ± SEM. *P*-values based on two-tailed Student's t-test. **n** Immunofluorescence staining showing hippocampal oligomeric and fibrillar Aβ. Scale bar = 10 μm.

serine residues (pS199 and pS396), but not threonine (pT231), in differentiated SH-SY5Y cells, whereas tau hyperphosphorylation on serine residues was prevented in the presence of SCH79797[48] (Fig. 10a–d; Supp. Fig. 4f–h;). No differences were observed in total tau protein levels, suggesting GrK has a specific effect on the signaling machinery leading to tau hyperphosphorylation (Supp. Fig. 4e, i).

To understand downstream signaling pathways induced by GrK, we analyzed the proteome of differentiated human neuroblastoma SH-SY5Y cells (treated with recombinant GrK or untreated) in the presence or absence of the PAR-1 inhibitor SCH79797 (Fig. 10e). We found that GrK significantly increased the abundance of 84 proteins, including proteins involved in the development of AD (Supporting Data 4). Pathway enrichment analysis applied to upregulated proteins revealed that the "Alzheimer's/Neurodegeneration" and "Kinase pathway" clusters were the most enriched (Fig. 10e). In the "Alzheimer's/Neurodegeneration" cluster, the most significantly enriched pathways included "amyloid fiber formation", "neurodegenerative diseases", and "deregulated CDK5 triggers multiple neurodegeneration" (Fig. 10f, Supporting Data 4). The "Kinase pathway" cluster of terms was characterized by the upregulation of pathways related to the activation of MAP kinases[49], which mediate tau hyperphosphorylation, and the "Post-translational protein phosphorylation pathway" (Fig. 10f). We also found that GrK induced the upregulation of "NF-κB signaling", which in neurons is associated with the modulation of ion channel protein expression, supporting Ca²⁺ increases at intraneuronal levels, and with a higher Aβ load[50]. Notably, the induction of all these pathways was prevented when GrK treatment took place in the presence of SCH79797, leading to the significant upregulation of only 12 proteins (Supporting Data 4), whereas enrichment analysis detected no pathways significantly related to these proteins. Together, these data clearly support our results in 3xTg-AD mice, suggesting that CD103⁻CD8⁺ T cells play a negative role in AD and that blocking GrK activity and its interaction with PAR-1 may have therapeutic value to reduce neuronal dysfunction and cognitive decline in AD.

## Discussion

CD8⁺ Trm cells form a defensive line against infections and cancer but may also play a role in the development of chronic inflammatory conditions[5]. Human and mouse brains contain CD103⁺CD8⁺ and CD103⁻CD8⁺ Trm cells[11,13,16,42,51], but their role in AD pathogenesis is unclear. Our scRNAseq analysis revealed an altered brain CD8⁺ T cell compartment with more abundant CD103⁻ Trm cells and the inhibition of signals promoting CD103 expression in the 3xTg-AD mouse model, which develops the diagnostic neuropathological features of human AD (amyloid and tau pathologies). We showed that CD103⁻CD8⁺ T cells originate from the circulation, use LFA-1 integrin to extravasate into the brain and selectively produce larger quantities of GrK in 3xTg-AD mice and human AD patients compared to controls. GrK induced neuronal dysfunction and tau hyperphosphorylation in mouse and

human cells via PAR-1, revealing a key immune mechanism that promotes neurotoxic inflammation and brain damage in AD (Supp. Fig. 6).

Previous studies in transgenic mice with separate Aβ or tau pathology showed the existence of CD8⁺ T cell subsets with protective or deleterious effects depending on the disease stage and pathological hallmarks[7,10–13,17]. Particularly, CD8⁺ T cells in APP/PS1 mice with advanced β-amyloid pathology show a transcriptomic profile similar to old WT animals and higher levels of CD103 were detected by flow cytometry[11]. Brain CD8⁺ T cells of 5xFAD mice with aggressive late-stage Aβ pathology had higher CD103 abundance together with higher expression of CXCR6 and PD-1 (compared to WT mice), inhibiting Aβ accumulation by suppressing the activation of microglia[13]. Previous studies using models of tau pathology suggested brain-infiltrating CD8⁺ T cells promote disease development, although additional signals appear to be essential for CD8⁺ T cell accumulation in the CNS of such mice[12,17]. Particularly, old P301S mice lacking APOE do not accumulate more brain CD8⁺ T cells despite tau pathology, suggesting that CD8⁺ T cell infiltration into the CNS requires APOE4[12]. In this context, our data show that the co-occurrence of Aβ and tau pathologies significantly reduces the abundance of intraparenchymal CD103⁺CD8⁺ Trm cells in the brains of 3xTg-AD mice compared to WT animals due to the downregulation of genes encoding transcription factors and cytokines involved in the differentiation and maintenance of CD103⁺ Trm cells[31]. Particularly, our results indicate strong suppression of the homeostatic and anti-inflammatory cytokine IL-33, which is required for the generation and survival of brain CD103⁺CD8⁺ Trm cells[52], and are in line with previous studies showing the loss of IL-33 in AD patients[53].

Our results also show that CD103⁻CD8⁺ Trm cells are more abundant in the brains of 3xTg-AD mice than WT controls. The accumulation of these cells (and the loss of CD103⁺ cells) was also observed in the brains of MS patients[54] and in primary Sjögren's syndrome[55] suggesting common dysregulation of the CD8⁺ T cell compartment in these diseases. Similarly, CD103⁻CD8⁺ Trm cells accumulated in the guts of patients with inflammatory bowel disease (IBD), and the rebalancing of CD103⁻CD8⁺ and CD103⁺CD8⁺ Trm cells was associated with disease remission[56], suggesting CD103⁻ cells promote disease pathogenesis. Indeed, our data demonstrate that CD103⁻CD8⁺ T cells are detrimental in 3xTg-AD mice and their depletion in the brain ameliorates neuropathology and cognitive deficits. Importantly, our data from AD patients show an increase in the abundance of intraparenchymal CD103⁻CD8⁺ T cells in the hippocampus while the percentage of CD103⁺CD8⁺ T cells declines, suggesting CD103⁻ cells may also promote disease development in human AD patients.

We also found that brain CD103⁻CD8⁺ T cells originate from the circulation in 3xTg-AD mice. Previous studies have shown that intratissutal cytotoxic CD103⁻CD8⁺ Trm cells originate from circulating CXCR3⁺ "exKLRG1" effector cells[26] and our single-cell data showed that *Cxcr3* was one of the best marker genes for the CD103⁻CD8⁺ Trm cell population, providing additional evidence that these cells migrate

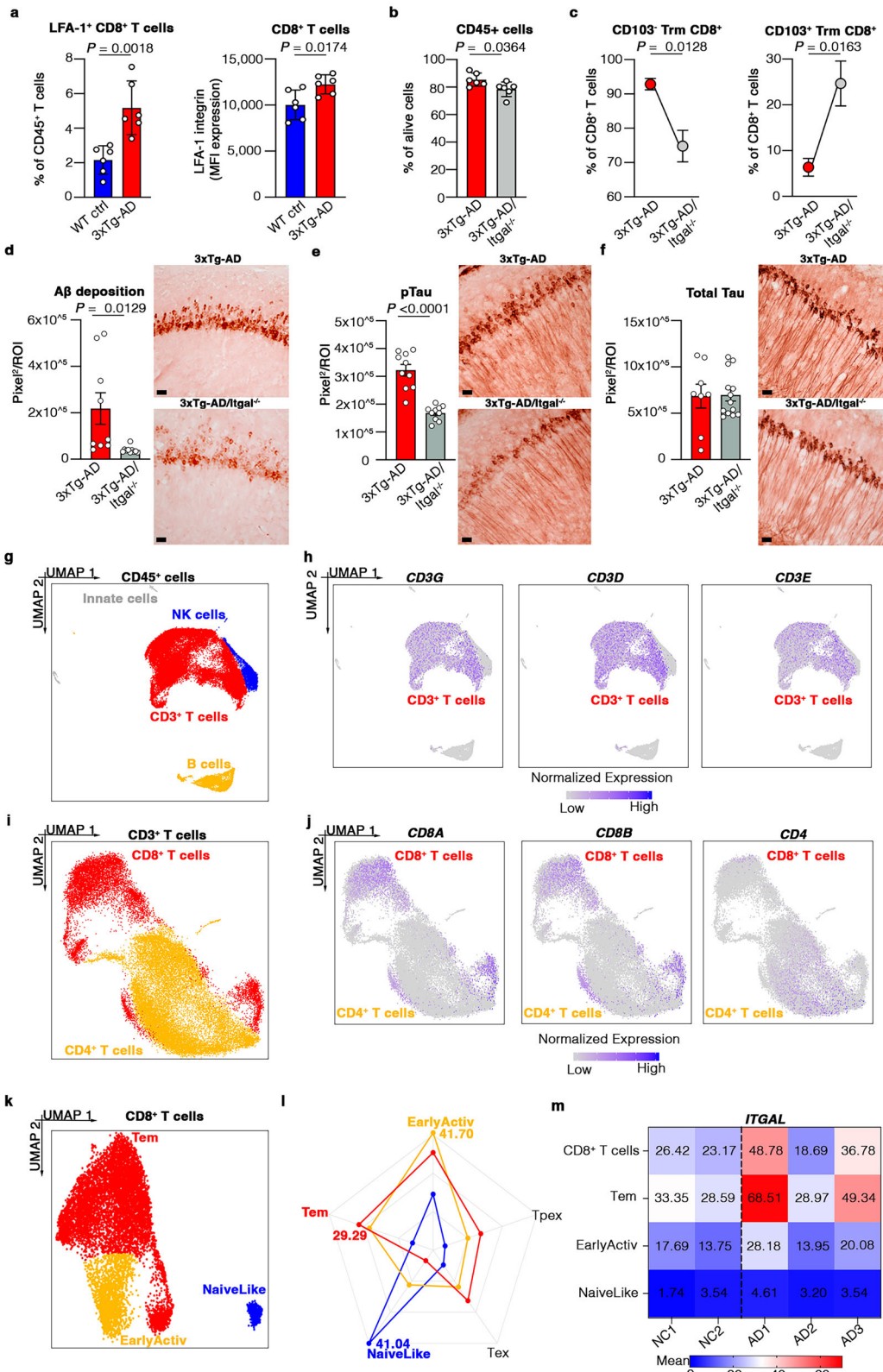

from the blood into the brain. CXCR3 was recently shown to regulate CD8[+] T cell infiltration and neuronal damage in a human 3D model of neuro-immune interactions, and CXCL10 (the chemokine ligand of CXCR3) accumulates in AD, suggesting that the trafficking of detrimental CD8[+] T cells in the AD brain is mediated by CXCR3–CXCL10 interaction[57]. Our data also show the massive depletion of CD103[−]CD8[+] T cells in the brains of 3xTg-AD/*Itgal*[−/−] mice lacking LFA-1, suggesting a

role for this integrin in the migration of detrimental CD8[+] T cells into the brain during AD. Interestingly, an LFA-1 blockade in 3xTg-AD mice reduces neutrophil migration into the CNS and mitigates disease symptoms, suggesting a key role for this integrin in the migration of circulating leukocytes into the AD brain[22]. Our results also showed that the percentage of circulating LFA-1[+]CD44[+]KLRG1[+]CD8[+] Tem cells increased (along with higher LFA-1 levels) in 3xTg-AD mice compared

**Fig. 6 | LFA-1 integrin controls the accumulation of CD8⁺CD103⁻ cells in the brains of 3xTg- AD mice. a** Flow cytometry showing the percentage of LFA-1⁺ CD8⁺ T cells (left) and corresponding MFIs (right) in the blood of WT ($n = 6$) and 3xTg-AD ($n = 6$) mice. Data are means ± SD, with $P$-values based on two-tailed Student's t-test. **b** Flow cytometry showing the percentage of CD45+ leukocytes among live cells. Data are means ± SEM, with $P$-values based on two-tailed Student's t-test. **c** Flow cytometry showing the percentage of CD103⁻ (left) and CD103⁺ (right) CD8⁺ Trm cells in the brains of 3xTg-AD ($n = 6$) and 3xTg-AD/$Itgal^{-/-}$ ($n = 6$) mice. Data are means ± SEM, with $P$-values based on two-tailed Student's t-test. **d–f** Representative immunohistochemical staining of the hippocampus in 3xTg-AD ($n = 3$) and 3xTg-AD/$Itgal^{-/-}$ ($n = 3$) mice, showing the Aβ load (c), and the levels of hyperphosphorylated (d) and total (e) tau protein. Scale bar = 20 μm. Data are means ± SEM, with $P$-values based on two-tailed Student's t-test. **g** UMAP plot showing immune cell populations in human blood ($n = 3$ AD patients and $n = 2$ negative controls, NCs) by scRNAseq. **h** Normalized expression of *CD3G*, *CD3D* and *CD3E* on a UMAP plot. Transcript levels are color- coded **i** CD3⁺ T cell cluster subsets: UMAP plot showing CD8⁺ and CD4⁺ T cell subpopulations in human blood ($n = 3$ AD patients and $n = 2$ NCs). **j** Normalized expression of *CD8A*, *CD8B* and *CD4* on a UMAP plot. Transcript levels are color- coded. **k** CD8⁺ T cell cluster subsets: UMAP plot showing early active, Tem, and naïve-like CD8⁺ T cells in human blood ($n = 3$ AD patients and $n = 2$ NCs). **l** Radar plot showing AUCell score using known genes. **m** Heat map reporting the expression values (mean calculated using the Hurdle model) of *ITGAL* in AD patients ($n = 3$) and NCs ($n = 2$). Transcript levels are color-coded.

to WT controls, suggesting that circulating CD8⁺ T cells are stickier and more prone to migrate into the AD brain. This is also supported by the higher expression of ICAM- 1, the vascular ligand of LFA-1, on the brain endothelial cells of AD mouse models and human AD patients, indicating that vascular inflammation may promote the recruitment of circulating activated CD8⁺ T cells into the brain. Finally, our analysis of a published dataset showed that *ITGAL* (encoding the CD11a subunit of LFA-1) was upregulated in the whole population of circulating CD8⁺ T cells in AD, suggesting that the LFA-1–ICAM-1 and CXCR3–CXCL10 axes may act in concert to promote the invasion of the brain by peripheral CD8⁺ T cells in AD.

We found that most CD103⁺CD8⁺ and CD103⁻CD8⁺ T cells expressed CD69, a marker of both "residency" and T cell activation, suggesting CNS Trm cells may have effector activities[5]. However, CD103⁺CD8⁺ and CD103⁻CD8⁺ T cells have distinct transcriptional phenotypes and functions in the normal intestine or during pathological conditions, including a model of brain viral infection[11,26,32,51,56]. The analysis of our datasets from 3xTg-AD and WT mice are consistent with these data and unbiased cell fate trajectory analysis clearly showed that brain CD8⁺ T cell differentiation occurred on a tightly organized trajectory starting from a common root and ending with two differentiation states distinguished by the expression of *Itgae* and *Eomes*. However, only CD103⁻ (not CD103⁺) CD8⁺ Trm cells infiltrating the brains of 3xTg-AD mice were strongly associated with the "*Alzheimer's disease*" pathway, suggesting these cells have distinct pathological functions specifically promoting AD. Importantly, CD103⁻CD8⁺ Trm cells accumulating in the brains of AD patients and 3xTg-AD mice were characterized by no significant increase in *Gzmb* and *Prf1* expression combined with the strong expression of *Gzmk* and high levels of GrK protein (but low levels of GrA and GrB proteins, a distinct phenotype for CD8⁺ Trm cells in AD. Indeed, recent studies performed in other contexts have shown that GrK is an atypical cytotoxic molecule that can be produced and released independently of GrA and GrB and can act in the absence of perforin[58,59], thus supporting the phenotype of the CD103⁻GrK⁺CD8⁺ T cells we found in AD. GrK⁺ T cells may respond to cytokine stimuli alone, given that TCR stimulation can downregulate *Gzmk* expression, suggesting that GrK⁺ T cells, despite their activated phenotype, are not antigen-stimulated T cells in AD[59].

Higher numbers of GrK⁺CD8⁺ T cells have been reported in several pathologies, including immune aging, cancer and autoimmune diseases[55,58–61], suggesting AD pathogenesis may share common immune mechanisms with other inflammatory conditions. In a heterogeneous group of subjects, GrK was more abundant in brain CD103⁻CD69⁺CD8⁺ T cells than CD103⁺CD69⁺CD8⁺ cells populations, but most brain donors had no neurological disease or were patients with MS or Parkinson's diseases and no data could be correlated with AD[16]. A recent study found an increased percentage of CD69⁺CD103⁺ cells in the CSF of a small group of AD patients, relative to the total population of memory CD45RA⁻CD8⁺ T cells[62]. However, these authors did not apply their CSF analysis to Trm cells (the majority of T cells in the brain parenchyma) and did not consider that CSF cells contain a significant population of KLF2⁺ cells with a

circulation origin, as shown by our data[62]. CD8⁺ T cells have also been found in the leptomeninges of AD patients[9], and despite recent studies showing the abundance of meningeal immune cells and the role of CNS borders during neurodegenerative and neuroinflammatory diseases[15], our results reveal that the number of CD103⁻CD8⁺ Trm cells does not change significantly and GrK secretion does not increase in the meninges, suggesting the brain is the main site of CD8⁺ Trm cell dysregulation.

How CD8⁺ T cells communicate with neuronal cells and potentially induce deleterious effects in AD is unclear. The accumulation of CD8⁺ T cells in the CNS of aged mice was recently found to promote axonal degeneration and age-related cognitive and motor decline through the release of GrB[14], which we did not find in our studies of 3xTg-AD mice. In addition, age-related alteration of brain functions has also been associated with clonally expanded CD8⁺ T cells producing INF-γ, which infiltrate old neurogenic niches, inhibiting the proliferation of neural stem cells[63]. The ablation of circulating CD8⁺ T cells in APP/PS1 mice does not reduce the Aβ load or cognitive decline but increases the expression of neuronal genes such as *Arc* and *Npas4*, suggesting that brain CD8⁺ T cells may modulate synaptic plasticity[10]. Nevertheless, CD8⁺ T cells were found close to neurons in AD patients, but their ability to exert direct neurotoxic functions has not been shown[9]. Our data fill this knowledge gap and demonstrate that CD103⁻CD8⁺ Trm cells induce neuronal dysfunction by releasing GrK. Indeed, our high-resolution live imaging studies showed that neurons from 3xTg-AD mice cultured in the presence of GrK⁺CD103⁻CD8⁺ Trm cells undergo profound functional changes. Notably, purified active GrK also induced neuronal dysfunction, confirming the neurotoxic role of this molecule. Our proteomic data on human neuronal cells further supported these observations, showing that GrK-PAR-1 interactions activate several intracellular pathways involved in neurodegeneration and AD. Moreover, we show that GrK⁺CD8⁺ T cells accumulated in the parenchyma near hippocampal neurons in AD patients and 3xTg-AD mice, whereas GrK⁻CD8⁺ T cells were preferentially located in the healthy hippocampus, further suggesting a neurotoxic role for GrK. Our data thus suggest that GrK is the basis of a key immune mechanism that induces brain neurotoxicity with a selective role in the mediation of CD103⁻CD8⁺ Trm cell-dependent neuronal changes in AD.

We found that GrK released by CD103⁻CD8⁺ Trm cells activates PAR-1 (the classical thrombin receptor) on primary AD neurons, leading to functional alterations and potentially affecting neuronal networks[64]. GrK may also activate PAR-1-expressing glial cells, such as astrocytes, which release glutamate following PAR-1 engagement, further contributing to neuronal alterations[65]. Previous studies have shown that GrK cleavage and PAR-1 activation also induce the phosphorylation of ERK1/2 and MAPK, triggering the secretion of proinflammatory cytokines such as IL-1β, MCP-1, IL-6, and IL-8, as well as upregulating the expression of adhesion molecules on monocytes, fibroblasts and cells of epithelial origin[35]. Considering these proinflammatory activities of GrK, and our data showing that the ablation

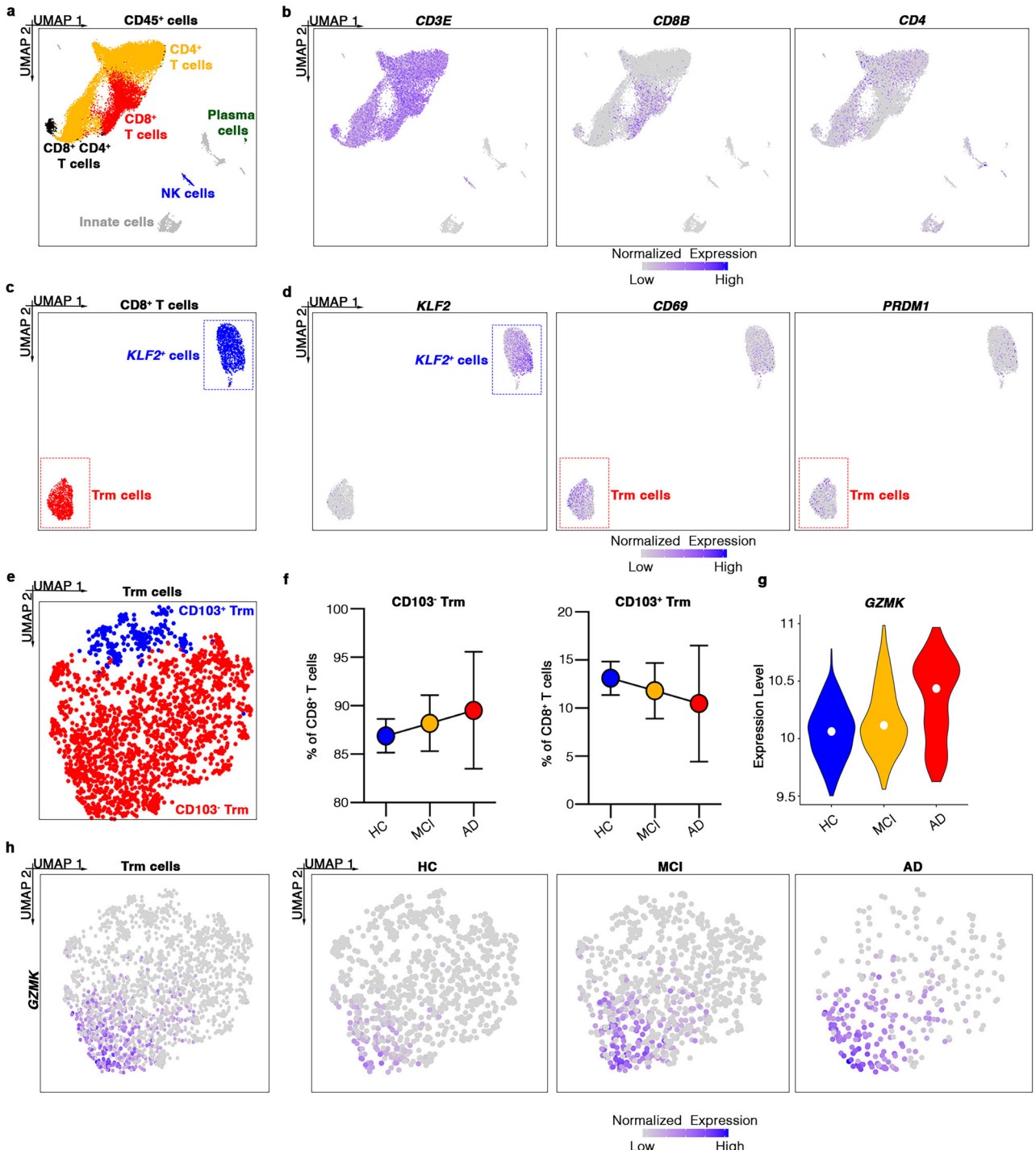

**Fig. 7 | GrK⁺CD103⁻CD8⁺ Trm cells accumulate in the CSF of AD patients. a** UMAP plot showing leukocyte populations detected in human CSF ($n = 4$ AD, $n = 5$ MCI, and $n = 9$ healthy controls, HCs) by scRNAseq. **b** Normalized expression of *CD3E, CD8B* and *CD4* in the UMAP plot. Transcript levels are color-coded. **c** CD8⁺ T cell cluster subsets: UMAP plot showing *KLF2*⁺ and Trm cell subpopulations in human CSF ($n = 4$ AD, $n = 5$ MCI, and $n = 9$ HCs). **d** Normalized expression of *KLF2, CD69* and *PRDM1* in a UMAP plot. Transcript levels are color-coded. **e** Trm cell cluster subsets: UMAP plot showing CD103⁻ and CD103⁺ Trm cells in human CSF ($n = 4$ AD, $n = 5$

MCI, and $n = 9$ HCs). **f** Percentage of CD103⁻ and CD103⁺ CD8⁺ cells in human CSF ($n = 4$ AD, $n = 5$ MCI, and $n = 9$ HCs). Data are means ± SD. Source data are provided as a Source Data file. **g** Violin plot reporting *GZMK* expression in Trm cells of $n = 4$ AD patients, $n = 5$ MCI subjects, and $n = 9$ HCs. The white dot represents median expression. **h** Normalized expression of *GZMK* in a UMAP plot of all Trm cells (panel 1) and Trm cells from HCs (panel 2), MCI subjects (panel 3), and AD patients (panel 4).

of brain CD103⁻CD8⁺ T cells (the majority expressing GrK) mitigates AD in mouse models, we propose that GrK–PAR-1 interactions promote chronic neuroinflammation and neurodegeneration in AD.

PAR-1 has a negative role in several neuropathological conditions and its inhibition reduces brain damage and inflammation in animal

models of brain ischemia[66]. Although not related to CD8⁺ T cells, PAR-1 inhibition also ameliorates cognitive performance in rats injected intracerebrally with Aβ peptide 1–42[40]. However, PAR-1 engagement can also promote learning and synaptic plasticity under physiological conditions[39]. Notably, our data show an increase in neuronal PAR-1

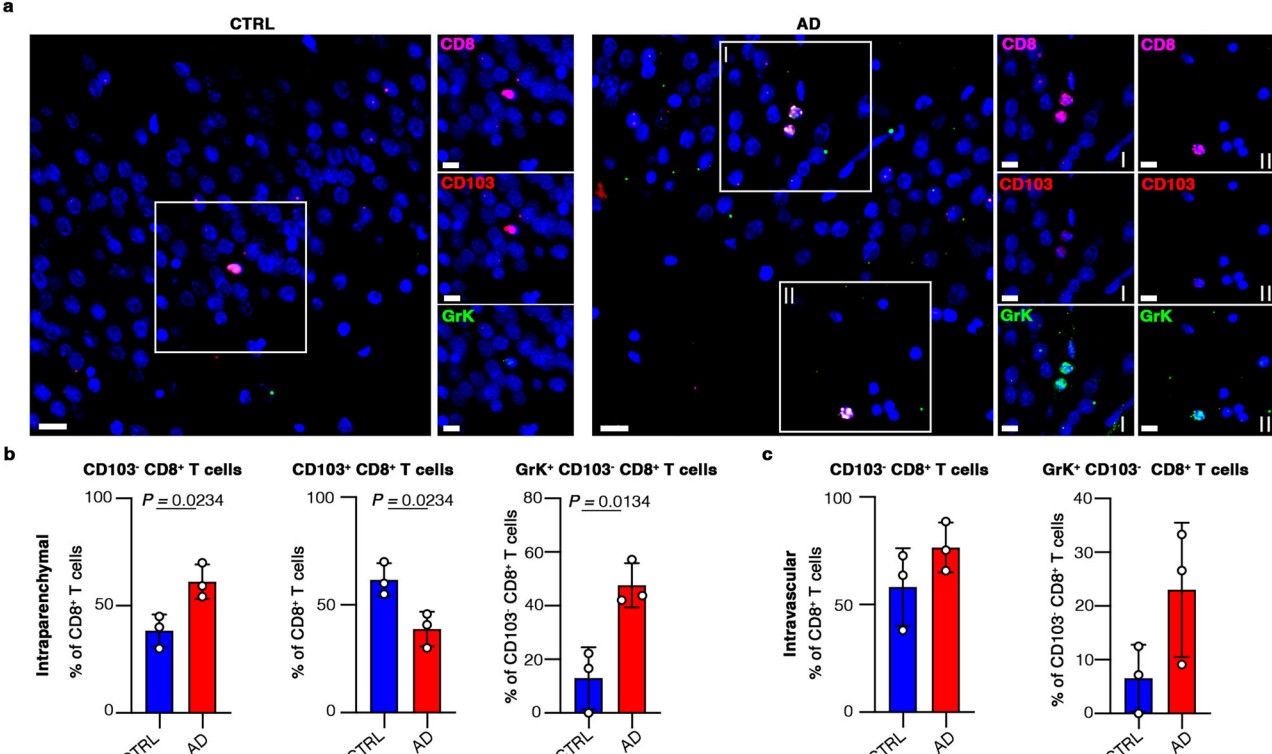

**Fig. 8 | GrK⁺CD103⁻CD8⁺ Trm cells accumulate in the brains of AD patients. a** Immunofluorescence staining of control (CTRL) and AD human brain tissues. Scale bar = 40 μm, or 15 μm in the zoomed images. **b** Bar plots showing the percentage of intraparenchymal CD103⁻, CD103⁺, and GrK⁺CD103⁻ CD8⁺ T cells in the hippocampus of HCs (*n* = 3) and AD patients (*n* = 3). Data are means ± SD (two-tailed Student's t-test). Source data are provided as a Source Data file. **c** Bar plots showing the percentage of intravascular CD103⁻and GrK⁺CD103⁻ CD8⁺ T cells in the hippocampus of HCs (*n* = 3) and AD patients (*n* = 3). Data are means ± SD (two-tailed Student's t-test). Source data are provided as a Source Data file.

expression in 3xTg-AD mice, suggesting that chronic alterations in PAR-1 signaling promote immune-driven neuropathology in AD.

Importantly, our data demonstrate that neuronal PAR-1 activation by GrK in mice and humans induces tau hyperphosphorylation in vitro[48] and these results are in line with previous studies showing that the number of CD8⁺ T cells in the brains of AD patients correlates with tau pathology and Braak staging[10,17]. Also, the percentage of T cells in the brains of mice with tau pathology was higher than that detected in animals with amyloidosis[12]. Systemic CD8⁺ T cell ablation has no effect on behavioral changes or neuropathology in APP-PS1 mice[10], further highlighting the pathological link between tau and CD8⁺ T cells. Along with our work, this evidence may explain why an increase in the abundance of CD103⁻CD8⁺ Trm cells was not observed in mouse models of amyloidosis[11,13]. However, we found that the ablation of brain CD103⁻CD8⁺ T cells (86% of which produce GrK) in 3xTg-AD mice reduced both amyloid and tau pathologies. The effect of CD8⁺ T cell inhibition on amyloidosis is supported by the fact that Aβ and tau pathologies have a synergistic effect in AD and tau pathology itself may favor Aβ deposition[19]. Together, these data indicate that CD8⁺ T cells can directly sustain tau pathology during AD, mechanistically linking the invasion of the brain by CD8⁺ T cells and neuroinflammation with the development of AD neuropathology.

In conclusion, we show that dysregulated CD103⁻CD8⁺ T cells and GrK−PAR-1 signaling contribute to changes in communication between the immune system and CNS, leading to neuroinflammation and neuronal dysfunction in AD. GrK inhibitors have already been developed to treat inflammatory diseases, including myocardial infarction[35], whereas drugs targeting PAR-1 are already used to block platelet aggregation[67], suggesting that the manipulation of GrK−PAR-1 interactions may also benefit AD patients.

## Methods
All chemicals, kits, and antibodies are listed in Supp. Table 2.

### Ethics statement
Formalin-fixed paraffin-embedded (FFPE) hippocampal sections of controls and Alzheimer's disease (AD) cases were obtained from the Medical Research Council (MRC) London Neurodegenerative Disease Brain Bank. All material was collected from donors (females aged 51-69 at time of death as listed in Supp. Table 1) from whom written informed consent for brain autopsy and the use of the material and clinical information for research purposes had been obtained by MRC London Neurodegenerative Disease Brain Bank, approved by the NRES Committee London − City & East (REC references: 08/H0704/128 + 5). Our reference number approved by the MRC London Neurodegenerative Disease Brain Bank is BDR TRID_240. The neuropathology studies on human brain samples were approved by the Ethical Committee from the University of Verona and Azienda Ospedaliera Universitaria Integrata (AOUI), in Verona (protocol nr. 20794).

### Mice
The research conducted in this study complies with all relevant ethical guidelines. Research involving animals was authorized by the Ethical Committee from the University of Verona and by the Italian Ministry of Health, Department of Veterinary Public Health, Nutrition and Food Safety, Directorate General of Animal Health and Veterinary Medicine (authorization no. 164/2016-PR and 876/2021- PR), as required by Italian legislation (D. Lgs 26/2014) as per the application of European Directive (2010/63/UE). All efforts were made to minimize the number of animals used and their suffering during the experimental procedures. 3xTg-AD (MMRRC stock no. 34830-JAX), *Itgal⁻/⁻* (stock no. 005257), and WT B6129SF2/J (stock no. 101045) mice

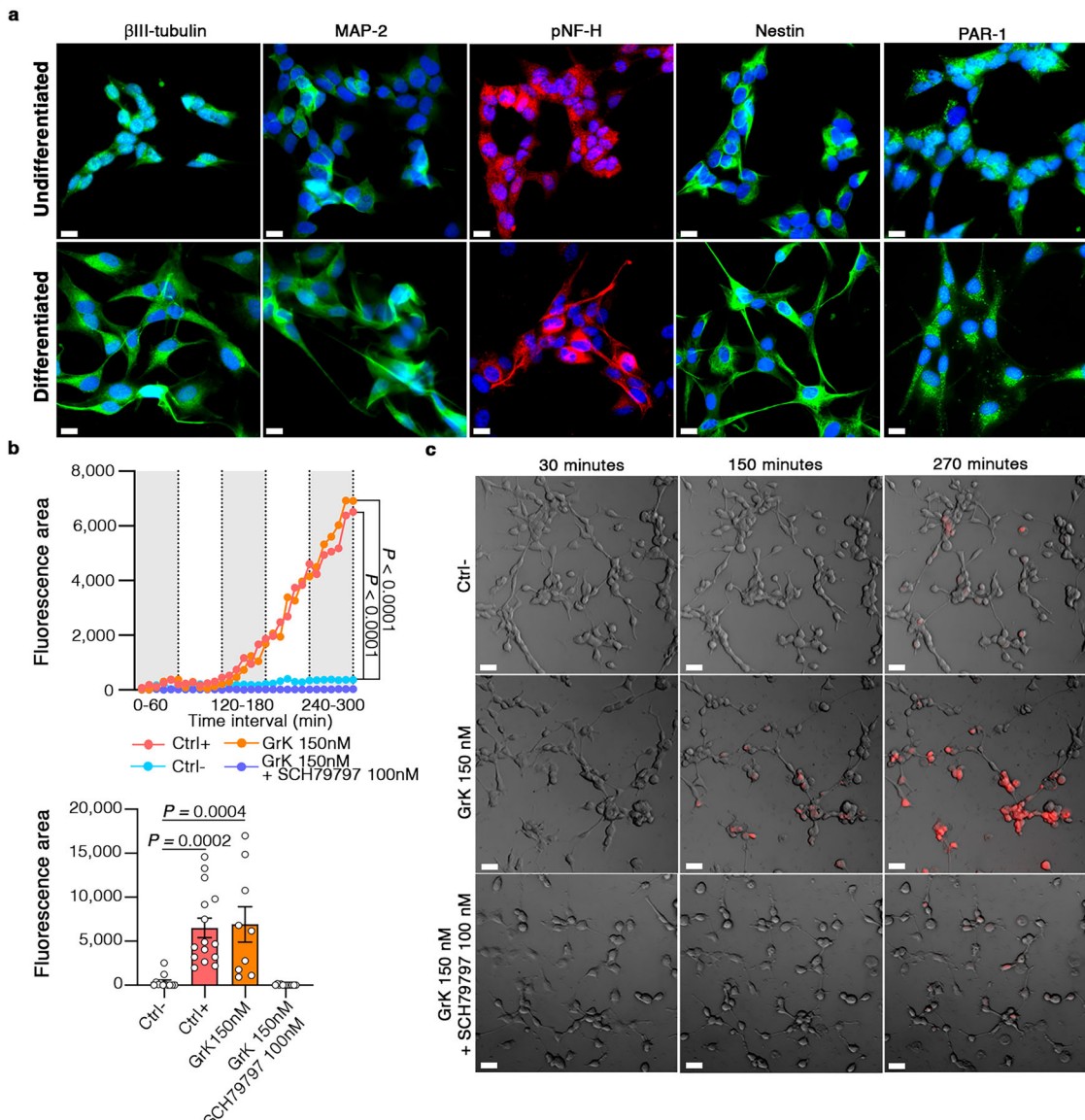

**Fig. 9 | GrK induces functional alterations in human neuronal cells via PAR-1.**
**a** Immunofluorescence staining for βIII-tubulin, MAP-2, pNF-H, Nestin and PAR-1 in undifferentiated and differentiated SH-SY5Y human neuroblastoma cells.
**b** Intracellular $Ca^{2+}$ release in differentiated SH-SY5Y cells cultured in the presence of 150 nM GrK, or 150 nM GrK and 100 nM SCH79797. The last time point is shown in the bottom panel. Ctrl⁻ = neurons alone. Ctrl⁺ = ionomycin-stimulated neurons (10 µM). Data are means ± SEM from three independent experiments. *P*-values based on two-way ANOVA-multiple comparisons. **c** Representative images of differentiated SH-SY5Y cells at different time points cultured without GrK (Ctrl⁻), in the presence of 150 nM active GrK, or in the presence of 150 nM active GrK plus 100 nM SCH79797. The red signal shows intracellular $Ca^{2+}$ release. Scale bar = 20 µm.

were purchased from the Jackson Laboratory. 3xTg-AD mice harbor human mutations for APP, PSEN,1 and TAU proteins, developing both amyloid and tau pathologies. We backcrossed 3xTg-AD and *Itgal*⁻/⁻ mice to obtain a transgenic line with all transgenes from the 3xTg-AD and *Itgal*⁻/⁻ models (APPSwe, tauP301L, PS1M146V knock-in and LFA-1 knockout). All mice were housed in pathogen-free climate-controlled facilities with 12 h dark/light cycle, and were provided with food and water *ad libitum*. For flow cytometry experiments, mice were anesthetized and perfused with $Ca^{2+}Mg^{2+}$ 1 mM in PBS 1X. For immuno-histochemical staining, mice were perfused with cold PBS 1X + $Ca^{2+}Mg^{2+}$ 1 mM and then fixed in 4% paraformaldehyde (PFA). For ablation experiments, depleting or isotype control antibodies were administered through an intraperitoneal injection every other day for 4 weeks. All mice were randomly assigned to the experimental groups. For each experiment, we used 6-month-old mice, both male and female.

## Preparation of single-cell suspensions
Before tissue collection, all mice were anesthetized by i.p. injection of ketamine (100 mg/kg body weight) and xylazine (15 mg/kg body weight) solution and perfused with $Ca^{2+}Mg^{2+}$ 1 mM in PBS 1X.

**Meninges.** Dura mater and leptomeninges were carefully removed from the interior aspect of the skull and surfaces of brain with fine surgical curved scissors and forceps, and enzymatically digested with a collagenase/DNase I solution (collagenase crude type IA, Merck Millipore; Deoxiribonuclease I crude lyophilized, Merck Millipore) at 37 °C for 15 min. Cells were washed with cold PBS 1X.

**Brain.** Brains were collected in cold PBS1X and separated from leptomeninges. After choroid plexus removal, brains were homogenized using a gentleMACS Dissociator (Miltenyi Biotec), and enzymatically

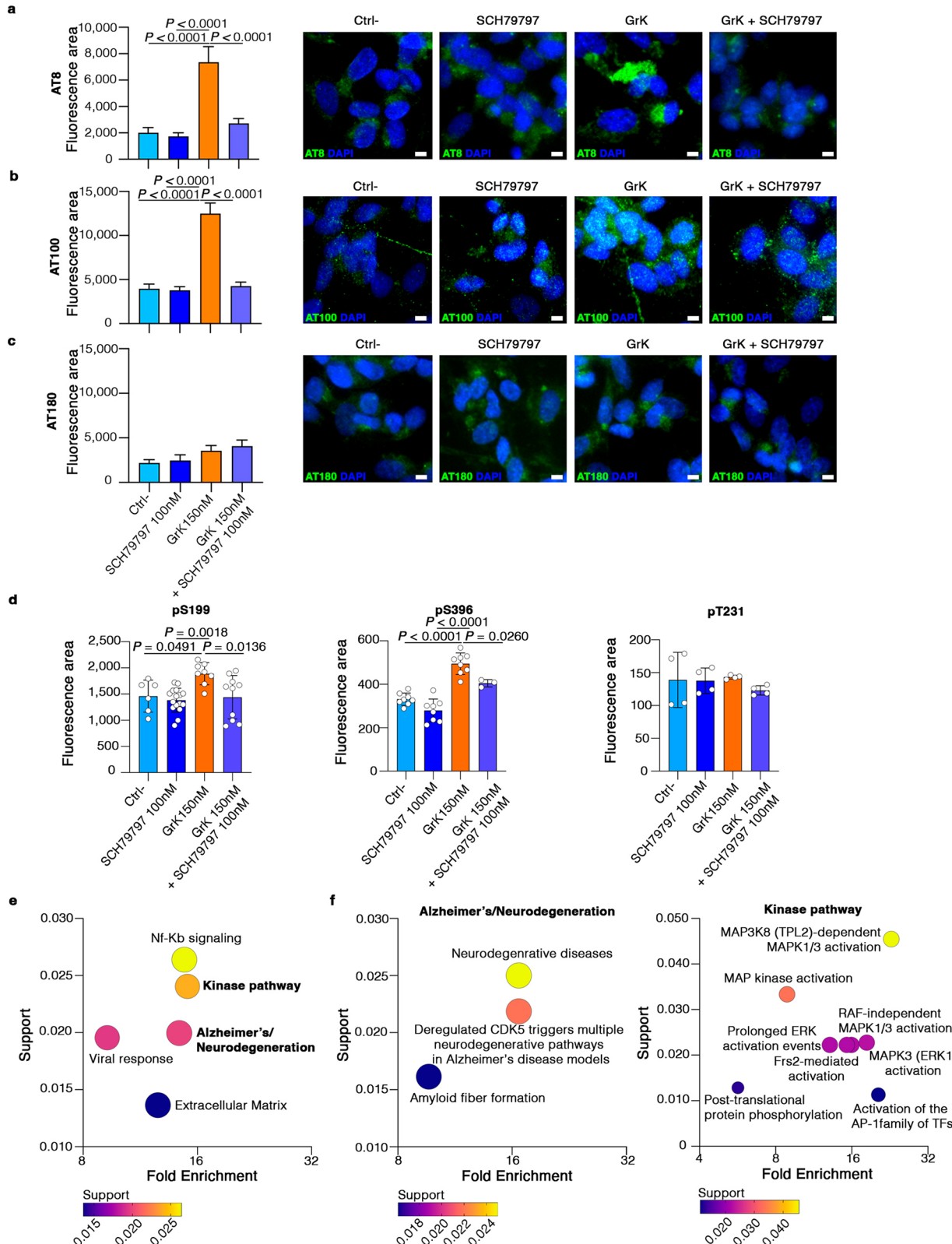

digested with a collagenase/DNase I solution (collagenase crude type IA, Merck Millipore; Deoxiribonuclease I crude lyophilized, Merck Millipore) at 37 °C for 45 min. Cells were passed through a 70-μm cell strainer into a new tube for Percoll (Merck Millipore) gradient centrifugation, and cells recovered from the interphase were washed with cold PBS 1X.

**Brain and meninges.** Brains were collected without peeling the meninges in cold PBS1X. After choroid plexus removal, brains were homogenized using a gentleMACS Dissociator (Miltenyi Biotec) and enzymatically digested with a collagenase/DNase I solution (collagenase crude type IA, Merck Millipore; Deoxiribonuclease I crude lyophilized, Merck Millipore) at 37 °C for 45 min. Cells were passed

**Fig. 10 | GrK induces tau hyperphosphorylation and alteration of signaling pathways in human neuronal cells via PAR-1. a–c** Bar plots and representative images showing tau phosphorylation levels on serine and threonine (AT8, AT100) (d, e) and only threonine (AT180) (f) residues in differentiated SH-SY5Y cells cultured alone (Ctrl⁻), in the presence of 100 nM SCH7979, in the presence of 150 nM active GrK alone, and the presence of both. Data are means ± SD of three independent experiments (two-way ANOVA-multiple comparisons). Scale bar = 5 μm. **d** ELISA showing the levels of phosphorylation on the pS199, pS396, and pT231 residues of tau protein in differentiated SH-SY5Y cells cultured alone (Ctrl⁻), in the presence of 100 nM SCH79797, in the presence of 150 nM active GrK alone, and in the presence of both. Data are means ± SD of three independent experiments. *P*-values are based on two-way ANOVA multiple comparisons. **e, f** Protein enrichment analysis from three independent proteomic experiments showing the best enriched clusters of terms (h) and terms belonging to the "Alzheimer's/neurodegeneration" and "kinase" pathways (i) in SH-SY5Y human neuroblastoma cells cultured in the absence of GrK and SCH79797 (Ctrl⁻), in the presence of 100 nM SCH79797 alone, in the presence of 150 nM active GrK alone, and in the presence of both. Source data are provided as a Source Data file.

through a 70-μm cell strainer into a new tube for Percoll (Merck Millipore) gradient centrifugation, and cells recovered from the interphase were washed with cold PBS 1X.

**Blood.** Blood samples were collected from the retro-orbital plexus of anesthetized mice using sodium heparinized capillaries and were mixed with an equal volume of 1% dextran from Leuconostoc spp (Sigma-Aldrich) plus 10 U/ml heparin. After pelleting the erythrocytes, the overlying supernatant plasma/dextran suspension of leukocytes was washed in cold PBS 1X.

**Spleen.** Spleens were mechanically disrupted, and single-cell suspensions were obtained by passing the cells through a 70-μm strainer. After erythrocyte lysis by NaCl 0.2% and 1.2%, cells were washed with cold PBS 1X.

**Liver.** Livers were collected in cold RPMI culture medium (Corning). They were mechanically homogenized firstly using a scalpel and then using a gentleMACS Dissociator (Miltenyi Biotec). Then, they were enzymatically digested with a collagenase/DNase I solution at 37 °C for 30 min. Cells were passed through a 70-μm cell strainer into a new tube for Percoll (Merk Millipore) gradient centrifugation, and cells recovered from the interphase were washed with cold PBS 1X.

### Single-cell RNA sequencing
**Sample preparation.** Meninges and brain were collected and homogenized as described above. Pools of eight female mice for 3xTg-AD and eight female mice for WT were used. The isolated leukocytes were washed with PBS 1X and labeled with an anti-CD45 BV480 antibody (1:20, BD Biosciences). After cell resuspension in PBS 1X with FBS10%, CD45⁺ and CD45^HIGH cells were sorted from meninges and brain, respectively. We used FACSAria Fusion device (BD) with BD FACSDiva software (8.0.1 BD). Both the 3xTg-AD and WT samples consisted of > 98% viable cells.

**Sequencing.** Sorted cells were resuspended to a final concentration of 700 cells/μl and cDNA sequencing libraries were prepared using the 10× Genomics Chromium Controller and the Chromium Single Cell 3′ GEM, Library and Gel Bead kit v3 (Pleasanton) following the manufacturer's instructions. Briefly, 10,000 live cells were loaded onto the Chromium Controller to recover 4000 single-cell gel-bead emulsions (GEMs) per inlet, uniquely barcoded. After cDNA synthesis, sequencing libraries were generated and final 10× library quality was assessed using the Fragment Analyzer High Sensitivity NGS kit (Agilent Technologies) before sequencing on the Illumina NextSeq500 platform, generating 75-bp paired-end reads (28 bp read 1, 91 bp read 2) at a depth of 50,000 reads per cell, yielding a median per-library depth of 72,783 reads per cell.

**Single-cell RNA-seq data cleaning.** Raw base call (BCL) files were processed using Cell Ranger v 6.0.1 (10× Genomics) to obtain a unique molecular identifier (UMI) count table for each sample. Briefly, two pipelines were used: *cellranger mkfastq*, which converts BCL files into FASTQ files, and *cellranger counts*, which takes FASTQ files from *cellranger mkfastq* and performs alignment, filtering, barcode counting,

and UMI counting. To perform these steps, *Mus musculus* reference data (version mm10 2020-A) were downloaded from the 10× official website.

**Normalization and analysis of Single-cell RNAseq data.** The SingleCellExperiment object was uploaded and analyzed using PartekFlow analysis software. First, we filtered out low-quality cells (doublets, damaged cells, or those with too few reads), evaluating the number of read counts per cell, the number of detected genes per cell, and the percentage of mitochondrial reads per cell. After the recommended normalization, through which counts were normalized and presented in logarithmic scale in CPM (count per million) approach, features were filtered, excluding genes that are not expressed by any cells in the dataset. Batch correction was provided using the Harmony task, available in the PartekFlow software. Principal component analysis (PCA) and Uniform Manifold Approximation and Projection (UMAP) dimensional reduction were performed before cell clustering based on AUCell task, which is a tool that permits to identify cells that are actively expressing genes within a gene list. We used published and available gene lists to annotate cells[14,46]. CD8⁺ T cells were then extracted and subclustered. Cluster biomarker genes were automatically computed by performing a Student's t-test on the selected attribute. Next, gene-specific analysis (GSA), pathway analysis, and gene set enrichment analysis (GSEA) were performed on the cell population of interest using PartekFlow plugins. We used PartekFlow software to run, after the scaling expression, the trajectory analysis, which is based on Monocle3 R package. GraphPad Prism 9.0 (v 9.0.0) was used to prepare donut, bubble, and bar plots. The following R-packages were used to prepare other graphs: Seurat (version 5.0.1), ggplot2 (version 3.4.4), hrbrthemes (version 0.8.0), viridis (version 0.6.5), monocle3 (version 1.3.1), ggpubr (version 0.6.0), ComplexHeatmap (version 2.15.4), Scillus (version 0.5.0).

**Analysis of a human Single cell RNAseq dataset.** We analyzed two publicly available datasets of Single-cell RNAseq data from AD patients. Those from Gate et al. harboring CSF leukocytes[9] (accessible under the accession code GSE134578), and the other harboring circulating leukocytes by Xu et al.[45] (accessible under the accession code GSE181279). We analyzed each of these datasets with PartekFlow software as previously described for our dataset, but using human references.

### Flow cytometry
**Surface staining.** After cell blocking with an anti-CD16/32 Fc-Block (1:100, BioLegend) for 10 min at room temperature (RT), cells were stained for 25 min in staining buffer (PBS1X with 10% Fetal Bovine Serum - FBS) at 4 °C in the dark. The following antibodies were used to stain: 1) Brain and meninges samples: CD11a/CD18 FITC (1:50, Miltenyi Biotech), CD103 BV421 (1:50, BD Biosciences), CD8 APC-H7 (1:50, BD Biosciences), CD69 BV650 (1:50, BD Biosciences), TCRγδ PE-CF594 (1:50, BD Biosciences), CD62L PE (1:50, BD Biosciences), CD45 BV605 (1:50, BD Biosciences), CD27 APC (1:50, BD Biosciences), CD11b APC-R700 (1:50, BD Biosciences), Ly6G BV510 (1:50, BD Biosciences), CD4 PE-Cy7 (1:50, BD Biosciences), CD197 BV786 (1:50, BD Biosciences), and CD44 BV711 (1:50, BD Biosciences); 2) Blood samples: CD11a/CD18 FITC (1:50, Miltenyi Biotech), CD62L PE (1:50, BD Biosciences), TCRγδ

PE-CF594 (1:50, BD Biosciences), VLA-4 PeCy7 (1:50, Miltenyi Biotech), Ly6G BV421 (1:50, BD Biosciences), CD44 BV510 (1:50, BD Biosciences), CD45 BV786 (1:50, BD Biosciences), CD4 APC (1:50, BD Biosciences), CD11b APC- R700 (1:50, BD Biosciences), and CD8 APC-H7 (1:50, BD Biosciences); 3) Spleen samples: CD11a/CD18 FITC (1:50, Miltenyi Biotech), CD103 BV421 (1:50, BD Biosciences), CD8 APC-H7 (1:50, BD Biosciences), CD69 BV650 (1:50, BD Biosciences), TCRγδ PE-CF594 (1:50, BD Biosciences), CD62L PE (1:50, BD Biosciences), CD45 BV605 (1:50, BD Biosciences), CD27 APC (1:50, BD Biosciences), CD11b APC-R700 (1:50, BD Biosciences), Ly6G BV510 (1:50, BD Biosciences), CD4 PE-Cy7 (1:50, BD Biosciences), CD197 BV786 (1:50, BD Biosciences), and CD44 BV711 (1:50, BD Biosciences); 4) Brain plus meninges: CD69 SB436 (1.5:50, ThermoFisherScientific), CD44 BV510 (1:50, BD Biosciences), CXCR3 (2:50, ThermoFisherScientific), TCRγδ BV650 (0.75:50, ThermoFisherScientific), CD45 BV711 (1:50, ThermoFisherScientific), CX3CR1 FITC (1.5:50, ThermoFisherScientific), CD103 PE (1.5:50, ThermoFisherScientific), CD62L (2.5:50, ThermoFisherScientific), CD4 PE-Cy7 (1.5:50, ThermoFisherScientific), CXCR6 APC (2.5:50, ThermoFisherScientific), CD8 APC-AF750 (4:50, ThermoFisherScientific). Antibodies against chemokine and cytokine receptors were added to the sample in a separate mix, and incubated 5 min 37 °C, and 10 min RT before to add other antibodies as previously described. Cells were then stained with 7AAD viability dye (BioLegend) for 5 min at RT before flow cytometry analysis.

**Intracellular staining.** To evaluate GrK expression, cells were stained with Viobility™ Fixable Dye (1:100, Miltenyi Biotec) for 15 min RT in the dark. After washing, cells were stained with CD45 APC-Vio770 (1:100, Miltenyi Biotec), CD8 PE-Cy7 (1:100, BioLegend), CD103 BV421 (1:100, BD Biosciences), CD3 BV650 (1:100, BD Biosciences) surface markers for 25 min at 4 °C in the dark. Next, cells were washed and resuspended firstly in Fixation buffer (BD Biosciences) and next in Intracellular Staining Perm Wash Buffer 1X (BD Biosciences). In the end, samples were stained with the primary anti-GrK antibody (1:100, PA550980, ThermoFisher Scientific) for 25 min at 4 °C in the dark, and then with the goat anti-rabbit AF488 secondary antibody (1:100, ThermoFisher Scientific) for 25 min at 4 °C in the dark. To evaluate expression of phenotypical markers, cells were stained with FVS700 (1:1000, BD Bioscience) in PBS 1X, and incubated for 15 min at 4 °C in the dark. After washing, and after cell blocking with an anti-CD16/32 Fc-Block (1:100, BioLegend) for 10 min at room temperature (RT), cells were stained with CD45 BV786 (2.5:50, ThermoFisher Scientific), TCRγδ BV650 (0.75:50, ThermoFisherScientific), CD4 PE-Cy7 (1.5:50, ThermoFisherScientific), CD8 APC-AF750 (4:50, ThermoFisherScientific), CD69 BV605 (1:50, BD Bioscience), CD103 BV711 (2.5:50, ThermoFisherScientific), CD44 BV510 (1:50, BD Biosciences), for 25 min at 4 °C in the dark. After washing, cells were fixed and permeabilized with Transcriptional Factor Buffer set (BD Pharmigen) following manufacturer's instructions. Then, samples were stained with the primary anti-GrK antibody (1:100, PA550980, ThermoFisher Scientific) for 25 min at 4 °C in the dark, and then with the goat anti-rabbit AF488 secondary antibody (1:100, ThermoFisher Scientific) for 25 min at 4 °C in the dark. After washing, cells were also stained with GrA eFluor 450 (1:50, ThermoFisher Scientific), GrB PE (1.75:50, SONY), and EOMES eFluor 660 (2.5:50, ThermoFisher Scientific) for 25 min at 4 °C in the dark.

**Analysis.** Flow cytometry analysis was performed using an LSR Fortessa X-20 flow cytometer (BD) with BD FACSDiva software (8.0.1 BD). Data were analyzed with FlowJo software (v 10). As gating strategy, we take cell of interest discriminating doublets. We counted as living cells those negative for 7AAD viability dye. Cells were gated for CD45 expression, identifying leukocytes. Next, we removed all the other cell subsets (Neutrophils Ly6g+CD11b+; Myeloid cells CD11b+; γδ T cells TCR γδ + ; CD4+ T cells) before detecting CD8+ T cells and analyzing the subset of interest (CD103- CD69+ and CD103+ CD69+ cells). Particularly,

we calculated the percentage of CD103- CD69+ and CD103+ CD69+ cells on the total population of CD8+ T cells. Of note, we tested the anti-GrK antibody before to proceed with the experiments. A GrK+ population was correctly detected in the stained sample compared to unstained and AF488 Fluorescence Minus One (FMO) control (Supp. Fig. 2d–f). Gating strategies from each experiment are available in Supp. Fig. 5a–h.

## Sorting of CD8+ T cells

Livers were collected from 3xTg-AD mice between 6 and 8 months of age. Leukocytes were isolated from the liver as described above. Then, CD8+ T cells were enriched by negative selection using a CD8α + T cell isolation kit (Miltenyi Biotech) following the manufacturer's instructions. After cell blocking by an anti-CD16/32 Fc-Block (1:100, BD Biosciences) for 10 min at RT, sample was stained 25 min in staining buffer (PBS1X with 10% FBS) at 4 °C with the following primary antibodies: CD103 BV421 (1:20, BD Biosciences), and CD8 APC-H7 (1:520, BD Biosciences), CD69 BV605 (1:20, BD Biosciences). Cells were then stained with 7AAD viability dye (BioLegend) for 5 min at RT before cell sorting. Hepatic CD69+CD103+CD8+ and CD69+CD103-CD8+ T cells were sorted using FACSAria Fusion device (BD) with BD FACSDiva software (8.0.1 BD) from the viable population (Supp. Fig. 5e). Next, freshly sorted CD8 + T cells were washed and seeded in RPMI medium (Corning) supplemented with 2.5% penicillin/streptomycin (Sigma- Aldrich), 1% Glutagro (Corning), 10% FBS, IL-2 (5 ng/ml, R&D systems) and IL-7 (5 ng/ml, R&D systems) overnight before in vitro co-cultures.

## In vitro cell cultures

**Murine primary neurons isolation and culture.** Hippocampal brain region was dissected from the brains of new-born 3xTg-AD mice (3 to 7 days old) and dissociated using the Adult Brain Dissociation kit (Miltenyi Biotec) and the gentleMACS Dissociator (Miltenyi Biotec) according to manufacturer's instructions. After removing debris and red blood cells, the sample was labeled with a non-neuronal biotin antibody cocktail (Miltenyi Biotec), and neurons were enriched by negative selection. Neurons were resuspended in Neurobasal medium (Gibco) with 2% B-27 supplement (Gibco), 1% Glutagro (Corning) and 1% penicillin/streptomycin (Sigma-Aldrich). We seeded 200.000 neurons/well in 48-well plates pre-coated with poly-D-lysine and laminin (Merk Millipore). Neurons were cultured for two weeks, and then half of the culture medium was replaced with fresh medium every other day. Purity of neurons was >95%.

**Human neuroblastoma SH-SY5Y cell line differentiation and culture.** Human neuroblastoma SH-SY5Y cells (94030304-CDNA-20UL, Sigma-Merck) were cultured in DMEM/F-12 (Biowest) complete (supplemented with 10% FBS, 1% Glutagro (Corning), Aldrich). For the differentiation cells were seeded for immunofluorescence at a density of $8 \times 10^3$ cells/well in 24 well plate on coverslips previously coated with ECMax gel (Sigma-Aldrich) and for the calcium imaging assay at a density of $4 \times 10^3$ cells/well in 48 well plate. Cells were exposed to differentiation medium 24 h after seeding. For neuronal differentiation, cells were cultured in DMEM/F-12 complete supplemented with 10 µM retinoic acid (Sigma-Aldrich) for 6 days followed by a 3-day differentiation step in neurobasal medium (Gicbo) enriched with 50 ng/ml recombinant human BDNF (rhBDNF, Peprotech), 2 mM db-cAMP (Sigma-Aldrich), 20 mM KCl, 2% B27 supplement (Gibco), and 1% Glutagro (Corning). The expression of neuronal markers was assessed by immunofluorescence as shown in Fig. 7a.

## Measurement of intracellular Ca²⁺ release by neurons

For all experiments, neurons cultured in the absence of CD8 + T cells were used as negative control (Ctrl-), whereas neurons treated with 10 µM ionomycin (Sigma-Aldrich) were used as positive control (Ctrl + ) during the time-lapse live imaging experiment.

**In vitro co-cultures of Trm CD8⁺ T cells with neurons.** Primary murine neurons were labeled with Biotracker 609 Red Ca²⁺ AM Dye (Merck Millipore) according to the manufacturer's instructions. In parallel, CD69⁺CD103⁺ and CD69⁺CD103⁻ Trm CD8⁺ T cells were labelled with CellTracker™ Blue CMAC Dye (1:250, ThermoFisher Scientific). Then, CD69⁺CD103⁺CD8⁺ and CD69⁺CD103⁻CD8⁺ T cells were seeded separately on neurons at a 1:4 ratio.

**In vitro co-cultures of neurons with recombinant GrK or CD103⁻ Trm CD8⁺ T cells in the presence of SCH79797 PAR-1 inhibitor.** Primary murine neurons were labeled with Biotracker 609 Red Ca²⁺ AM Dye (Merck Millipore) according to the manufacturer's instructions and resuspended in phenol red-free medium in the presence or absence of SCH79797 PAR-1 inhibitor (DBA). Then, we add mouse recombinant GrK (Cusabio) or sorted CD69⁺CD103⁻ Trm CD8⁺ T cells stained with CellTracker™ Blue CMAC Dye (1:250, ThermoFisher Scientific) at a 1:4 ratio.

**In vitro cultures of differentiated human SH-SY5Y neuroblastoma cell line with GrK in the presence of SCH79797 PAR-1 inhibitor.** Differentiated human SH-SY5Y neuroblastoma cells were labeled with Biotracker 609 Red Ca²⁺ AM Dye (Merck Millipore) according to the manufacturer's instructions and resuspended in phenol red- free medium in the presence or absence of SCH79797 PAR-1 inhibitor (DBA). Then, human recombinant GrK (Cusabio) was added.

**Time-lapse live imaging acquisition and analysis.** Ca²⁺-dependent changes of intracellular fluorescence were acquired with a LD Plan-Neofluar 20×/0.4 Corr M27 objective mounted on AxioObserver 7 microscope (Zeiss), equipped with a thermostatic chamber and a Hamamatsu camera. Exposure time was manually set and left unmodified through time-lapse experiments. Images were acquired every 10 min for 5 h under controlled conditions. For each scene, time-lapse videos were split into individual frames and analyzed using ZEN v2.6 software (Zeiss). Data were expressed as positive area for intracellular Ca²⁺ staining overtime.

### Depletion of circulating CD8⁺ T cells

Female and male 3xTg-AD and WT mice of 6 months of age were treated using an anti-CD8α depleting antibody (0.22 mg/mouse, BioXCell) for a total of four weeks. Treatment was performed every other day through an intraperitoneal injection. Control mice were treated with an isotype antibody (produced in-house, Clone Y13-259). Blood was drawn as previously described to confirm ablation of circulating CD8⁺ T cells (Supp. Fig. 3a, b). After the treatment, mice were left untouched for other 4 weeks, to minimize the stress induced by the manipulation, before to proceed with behavioral tests.

**Behavioral tests.** Morris water maze (MWM) test and the Contextual fear conditioning (CFC) task were performed as previously described. Before MWM and CFC, we evaluate the basic abilities and anxiety of the mice through ledge, hindlimb clasping, and open field tests. Ledge consists in placing mice on cage's ledge, where they typically walk along, attempting to descend back to the cage. A scoring system is adopted to detect and evaluate coordination deficits, and the test was conducted in triplicate[68]. During the hindlimb clasping test, mice lifted, and the hindlimb position was observed for 10 seconds. Depending on hindlimb retraction, a score is given[68]. Open field test evaluates locomotion, exploration, and anxiety of mice (Supp. Fig. 3f). Mice were placed into the open field square cage (50x50x30 cm) and were left free to explore the environment for 20 min: 10 minutes of acclimatization and 10 minutes of test. Videos were analyzed by AnyMaze software.

**Neuropathological analyses of mouse brain tissues.** After behavioral tests mice were anesthetized and perfused as previously described.

Brains were collected and placed in 4% PFA overnight. Next, brains were transferred in 30% sucrose for cryoprotection. When fully soaked with sucrose, brains were frozen, protected by tissue-tek optimal cutting temperature (OCT, DDK Italia) embedded compound. Then, they were cut in 30 μm slices using a cryostat (CM1520 Leica). Coronal sections, after incubation with blocking buffer (2% Normal Goat Serum, plus 0.4% Triton X-100 in PBS 1X) for 1 hour, were stained with anti-Aβ (6e10, 1:1000 BioLegend) antibody, and with anti-pTau (AT180, 1:200 ThermoFisher Scientific), and anti-total tau (HT7, 1:200 ThermoFisher Scientific) antibodies. Particularly, for tau staining the sections were treated with citrate buffer pH = 6 (BioOptica) at 85 °C for 30 min for antigen retrieval. After washing with PBS 1X plus 0.05% Tween-20, sections were added in 3% H₂O₂ for 10 min at RT, followed by the incubation with the biotinylated goat anti-mouse secondary antibodies for 2 h at RT. The immunoreactivity was visualized using DAB reagent. Images were acquired with Plan-Apochromat 20×/0.8 M27 objective on Axio Imager.Z2 microscope equipped with Axiocam 506 Color (Zeiss). Images were analyzed with ZEN blue software. Brightness, contrast, and color balance were adjusted over the whole image without eliminating any information present in the original. Investigators were blinded with respect to the genotype of the mice and the treatment. We analyzed a minimum of five sections per mouse.

### Immunohistochemical staining on paraffin-embedded mouse tissues and image analysis

Mice were anesthetized and perfused as previously described. After overnight fixation with 4% PFA, brains were washed and, after following treatments with ethanol 70%, ethanol 90%, ethanol 100%, and xylene, they underwent paraffin-embedding. 5 μm thick coronal brain slices were obtained using a microtome (DiaPath). Following treatments with xylene, ethanol 100%, ethanol 90%, and ethanol 70%, sodium citrate was used for antigen retrieval. Sections were incubated for 2 h with blocking buffer (2% Normal Goat Serum, plus 0.5% Triton X-100 and 2,5% Fetal Bovine Serum in PBS 1X) before being stained with rabbit anti-mouse PAR-1 antibody PAR-1 (1:100, Bioss). After washing with PBS 1X plus 0.05% Tween-20, sections were added in 3% H2O2 for 10 min at RT, followed by incubation with the biotinylated goat anti-mouse secondary antibodies for 2 h at RT. The immunoreactivity was visualized using DAB reagent. Images were acquired with Plan- Apochromat 20×/0.8 M27 objective on Axio Imager.Z2 microscope equipped with Axiocam 506 Color (Zeiss). Images were analyzed with ZEN blue software. Brightness, contrast, and color balance were adjusted over the whole image without eliminating any information present in the original. Investigators were blinded with respect to the genotype of the mice. We analyzed a minimum of nine sections per mouse.

### Preparation of brain homogenates and cell lysates

**Brain homogenates.** Brain homogenates were prepared as described by Illouz T. et al.[69]. Briefly, mice were perfused as previously described, and half brains were snap frozen and stored at −80 °C. Then, half brains were weighted, cut in small pieces, pottered, and sonicated with a volume of RIPA lysis extraction buffer (Thermo Fisher Scientific) dependent on their weight. Then, samples were centrifuged at 17,000× g for 90 min. Supernatants were stored as soluble fraction, while pellet was resuspended in tri-fluoro- acetic acid (TFA) and then dry under N2 gas stream. After resuspension in PBS 1X and neutralization with NaOH, samples were centrifuged at 17,000 g for other 90 min. Supernatants were stored as insoluble fraction. Soluble and insoluble fractions were snap frozen and stored at −80 °C.

**Cell lysates.** SH-SY5Y cells were growth, differentiated, and stimulated with recombinant GrK in the presence or absence of SCH79797 PAR-1 inhibitor. Unstimulated cells were used as negative control. Then, cells were washed with PBS 1X and then lysed using 200 μl of RIPA lysis

extraction buffer (Thermo Fisher Scientific). Samples were incubated for 10 min and then centrifugated at 17,000×*g* for 10 min. Supernatants were stored at −80 °C.

**ELISA.** Proteins present in brain homogenates (soluble and insoluble fraction) or cell lysates prepared as described above were measured using Bradford assay (SERVA) following the manufacturer's instructions. ELISA experiments were performed loading 2 μg/μl of proteins for brain homogenates soluble fractions, 0.2 μg/μl of proteins for brain homogenates insoluble fractions, and 0.05 μg/μl of proteins for cell lysates. 1-40 Aβ and 1-42 Aβ, or tau pT231, pS199, pS396 or total tau were measured by ELISA following manufacturer's instructions.

**Dot blot.** Proteins from brain homogenates (soluble and insoluble fraction) prepared as described above were measured using Bradford assay (SERVA) following the manufacturer's instructions. Dot blot experiments were performed, loading 200 μg/μl of proteins for the soluble fractions and 40 μg/μl of proteins for the insoluble fractions. After sample adsorption on the blotting membrane (Amersham), we performed blocking with 5% milk in TBS 1X. Next, membranes were stained with anti-oligomers (A11, ThermoFisher Scientific) or anti-fibrils (OC, ThermoFisher Scientific) antibodies for 1 h RT. After washing, membranes were stained with HRP secondary antibody for 1 h RT. After washing, membranes were developed using ECL (MerckMillipore).

### Immunofluorescence staining

For immunohistochemical staining mice were anesthetized by i.p. injection of ketamine (100 mg/kg body weight) and xylazine (15 mg/kg body weight) solution and perfused with cold PBS 1X + Ca$^{2+}$Mg$^{2+}$ 1 mM and then fixed in 4% paraformaldehyde (PFA).

**Frozen mouse tissue.** Free-floating PFA-fixed coronal mouse brains sections were incubated in blocking buffer (2% Normal Goat Serum, plus 0.4% Triton X-100 in PBS 1X) for 2 h at room temperature and then treated with rabbit anti-mouse CD8 (1:400, CellSignaling) primary antibody overnight at 4 °C, and then with goat anti-rabbit AF594 secondary antibody (1:1000, Invitrogen) for 1 hour at RT in the dark. After washing with PBS 1X plus 0.05% Tween-20, we added rabbit anti-mouse GrK (1:500, ThermoFisher Scientific) overnight at 4 °C, and then, after washing, biotinylated goat anti-rabbit (1:500, Merck Millipore) for 1 hour RT. We wash with PBS 1X plus 0.05% Tween-20 before adding anti-biotin streptavidin AF488 (1:1000, Invitrogen) and we incubate 45 min at RT. After washing with PBS 1X plus 0.05% Tween-20, we added mouse anti-mouse NeuN (1:200, Sigma-Aldrich) overnight at 4 °C, and then rabbit anti-mouse AF680 (1:500, Invitrogen). Nuclei were stained with DAPI (1:1000, D9542, Sigma-Aldrich). In the end, the sections were washed with PBS 1X, transferred to glass slides, and mounted with Dako medium. Images were acquired with Plan- Apochromat 40×/0.95 M27 objective on Axio Imager.Z2 microscope equipped with a Hamamatsu camera. For imaging of fibrillar/oligomeric amyloid beta forms, brain slices were incubated with a blocking solution and then treated overnight at 4 °C with either rabbit polyclonal antibodies against amyloid oligomers (A11, 57006, ThermoFisher, 1:1000) or amyloid fibrils (OC, 57005, ThermoFisher, 1:1000) and anti-β-amyloid, 1-16 antibody (6E10, 80300, Biolegend, 1:1000). Then, slices were incubated with secondary antibodies: goat anti-rabbit AF488 (a11034, Invitrogen, 1:1000) and goat anti-mouse AF680 (A-21057 Invitrogen, 1:1000), for 1 hour at RT in the dark. After washing with PBS 1X containing 0.05% Tween-20, the sections were incubated with a pan-neuronal marker (MAB2300, Millipore, 1:200) and then rabbit anti-mouse AF568 secondary antibody (a10037, 1:1000, Invitrogen). Nuclei were stained with DAPI (1:1000, D9542, Sigma-Aldrich). Finally, the sections were transferred to glass slides and mounted with Dako medium. Z-stack images were acquired with

Plan- Apochromat 20×/0.8 M27 objective on Axio Imager.Z2 microscope equipped with Apotome.2 (Zeiss) and a Hamamatsu camera. The images were post-processed with Imaris (9.6.0 version), applying a Gaussian filter.

**Trm CD69$^+$ CD8$^+$ T cell suspension.** Hepatic CD69$^+$CD103$^-$CD8$^+$ (200.000 cells) and CD69$^+$CD103$^+$CD8$^+$ (200.000 cells) T lymphocytes, isolated as previously described, were fixed with 4% PFA for 20 min at RT. Fixed cells were washed and resuspended in blocking solution (2% Normal Goat Serum, 2% Bovine Serum Albumin, 0.2% Triton X-100) for 1 hour at 4 °C. Cells were stained firstly with anti-mouse GrK (1:250, ThermoFisher Scientific) primary antibody for 1 hour at RT, and then with goat anti-rabbit AF647 secondary antibody (1:1000, Invitrogen) for 1 hour at 4 °C in the dark. Nuclei were counterstained with DAPI (1:2000, Sigma-Aldrich). Cells were transferred on a poly-lysinated glass slide and left adhering for 20 min. Slides were mounted and acquired under the microscope Axio Imager.Z2 microscope (Zeiss) with a Plan Apochromat 100×/1.46 oil DIC (UV) M27 objective equipped with Hamamatsu camera.

**Primary murine neurons co-cultured in vitro with CD69$^+$ CD103$^-$ CD8$^+$ T cells.** Primary murine neurons isolated as previously described were co-cultured with 5 h with hepatic CD69$^+$CD103$^-$CD8$^+$ T lymphocytes (1:4 ratio), isolated as previously described. Next, cells were fixed with 4% PFA for 10 min at RT. Fixed cells were washed and resuspended in blocking solution (2.5% FBS, and 0.1% Triton X-100) for 30 min at RT. Cells were stained firstly with an anti-mouse GrK (1:500, ThermoFisher Scientific) primary antibody for 1 hour at RT, and then with goat anti-rabbit AF488 secondary antibody (1:1000, ThermoFisher Scientific) for 1 hour at RT in the dark. Then, cells were stained with anti-mouse PAR-1 (1:200, Bioss) for 1 hour at RT in the dark, and, after washing, with goat anti-rabbit AF546 secondary antibody (1:1000, Invitrogen) for 1 hour in the dark. Nuclei were counterstained with DAPI (1:2000, Sigma-Aldrich). Slides were acquired under the microscope Axio Imager.Z2 microscope (Zeiss) with a Plan Apochromat 100×/1.46 oil DIC (UV) M27 objective equipped with Hamamatsu camera. Wild-field imaging with dichroic filter was used to visualize the morphology of neurons and of CD8$^+$ T cells.

**In vitro differentiated human SH-SY5Y neuroblastoma cell line.** SH-SY5Y cells (both differentiated and undifferentiated) were fixed in 4% paraformaldehyde for 10 min at RT and then washed with PBS 1X. To verify the differentiation state of SH-SY5Y cells, fixed cells were permeabilized and blocked with 0.3% Triton X-100 + 10% Normal Goat Serum (Vector Laboratories) in PBS for 1 hour at RT. Then, cells were incubated overnight with the following primary antibodies at 4 °C: rabbit anti-human MAP2 (1:100, ThermoFisher Scientific), mouse anti-human Nestin (1:100, Sigma-Aldrich), rabbit anti-mouse βIII-tubulin (1:100, Cusabio), or rabbit anti-human NF-H (1:100, Abcam). Appropriate secondary antibodies conjugated with AF488 (1:800, Molecular Probes) or AF594 dyes (1:800, ThermoFisher Scientific) were used for 1 hour at RT. Nuclei were stained with DAPI (1:2000, Sigma-Aldrich). To verify GrK-dependent tau hyperphosphorylation, we stimulated differentiated SH-SY5Y cells untreated or pre-treated for 1 hour with SCH79797 PAR-1 inhibitor (DBA) with mouse recombinant GrK (Cusabio) for 24 h. Not treated differentiated SH-SY5Y cells were used as a negative control. Differentiated SH-SY5Y cells pre-treated for 1 hour with SCH79797 PAR-1 inhibitor (DBA) were used to ensure that SCH79797 PAR-1 inhibitor alone was unable to induce tau hyperphosphorylation in these cells. Next, cells were fixed in 4% PFA for 10 min. Fixed cells were permeabilized and blocked with 0.2% Triton X-100 + 4% BSA + 2% Normal Goat Serum (Vector Laboratories). Cells were stained with (i) mouse α- human HT7 primary antibody (1:1000, ThermoFisher Scientific) to visualize total tau protein, or (ii) mouse α-human AT8 primary antibody (1:1000, ThermoFisher Scientific) to

visualize tau hyperphosphorylation on serine 202 and threonine 205 residues, or (iii) mouse α-human AT100 primary antibody (1:1000, ThermoFisher Scientific) to visualize tau hyperphosphorylation on serine 214 and threonine 212 residues, or (iv) mouse α-human AT180 primary antibody (1:1000, ThermoFisher Scientific) to visualize tau hyperphosphorylation on threonine 231 residues, for 1 hour at RT in blocking solution (0.2% Triton X-100 + 4% BSA + 2% Normal Goat Serum). Next, cells were washed before adding goat anti-mouse AF488 secondary antibody (1:1000, Invitrogen) for 1 hour at RT. Nuclei were stained with DAPI (1:2000, Sigma-Aldrich). Slides were acquired under the microscope Axio Imager.Z2 microscope (Zeiss) with a Plan Apochromat 100×/1.46 oil DIC (UV) M27 objective equipped with Hamamatsu camera. Levels of total tau and hyperphosphorylated tau protein were analyzed with ZEN blue software. Brightness, contrast, and color balance were adjusted over the whole image without eliminating any information present in the original. Investigators were blinded with respect to the treatment. We analyzed a minimum of 10 regions of interest (ROIs) for each condition and we repeated the experiment three times.

### Immunofluorescence staining of human brain tissues and quantification

FFPE sections were deparaffinized and incubated with antigen retrieval solution. Sections will be treated with primary antibodies: rabbit anti-human CD8α antibody (1:100, Abcam), rabbit anti-human CD103 antibody (1:100, Abcam), and anti-GrK antibody (1:100, ThermoFisher Scientific). The sections were incubated sequentially overnight at 4 °C in blocking solution (5% BSA, 2% Normal Goat Serum, and 0.5% Triton x-100) and then were incubated with appropriate fluorophore-conjugated secondary antibodies. Nuclei were stained with DAPI (1:1000, Sigma-Aldrich). Finally, the sections were mounted with Dako medium. Images were acquired using a Zeiss Axio Imager.Z2 microscope (Zeiss) and manually quantified in blind fashion.

### Proteomic analysis

**Sample preparation for shotgun analysis.** Human neuroblastoma SH-SY5Y cells were growth, differentiated, and stimulated with GrK in the presence and absence of SCH79797 PAR-1 inhibitor as previously described. Next, cell extracts were prepared using the EasyPep Mini MS Sample Prep Kit (Thermo Fisher Scientific) following the manufacturer's protocol. Briefly, $1 \times 10^6$ cells per well were lysed in lysis buffer with Universal Nuclease by pipetting ten times. Protein concentration was measured using a Microplate BCA Protein Assay Kit (Pierce™, Thermo Fisher Scientific). 50 µg of protein were reduced, alkylated (50 µL solution, 95 °C, 10 min), then digested with 50 µL trypsin/Lys-C at 37 °C for 3 h. Digestion was stopped with 50 µL of Digestion Stop Solution. For peptide cleanup, samples were loaded onto peptide cleanup columns, centrifuged twice at 1500 × g for 2 min, washed with 300 µL of Wash Solution A and B, and eluted with 300 µL of Elution Solution. Peptides were dried in a vacuum centrifuge and resuspended in 0.1% TFA for LC-MS/MS analysis.

**Mass spectrometry analysis.** Liquid chromatography-mass spectrometry (LC-MS) analyses were performed using an Ultimate 3000 nano-UHPLC system coupled to an Orbitrap Fusion Lumos Tribrid mass spectrometer (Thermo Fisher Scientific). Tryptic peptides (1 µg per sample) were injected onto the column (2 µm, 500 × 0.075 mm) and initially washed for 5 min at a flow rate of 300 nL/min with 4% acetonitrile (ACN) in 0.1% formic acid (FA). Peptide separation was achieved using a linear gradient from 4% to 50% ACN over 90 min. Ionization was performed via a nanospray electrospray ionization (ESI) source in positive ion mode, with an applied voltage of 1.5 kV and a capillary temperature maintained at 275 °C. Mass spectrometry data acquisition was conducted in data-dependent acquisition (DDA) mode. Full MS1 spectra were collected in the Orbitrap analyzer over an m/z range of 375–1500 with a resolution of 120,000 (at m/z 200), standard automated gain control (AGC) and maximum injection time settings, and precursor ion charge states of 2+ to 5 +. For MS2 analysis, precursor ions exceeding an intensity threshold of 3.0 × 104 and within a charge range of 2+ to 5+ were selected within a 4.0 Da isolation window and fragmented using high-energy C-trap dissociation (HCD) at a normalized collision energy of 30%. The resulting fragment ions were analyzed in the Orbitrap at a resolution of 30,000 (at m/z 200). To minimize redundant fragmentation, a dynamic exclusion time of 45 s was applied. Each experimental condition was analyzed in triplicate.

To ensure data quality, blank samples (30% ACN in water) and a quality control (QC) sample consisting of HeLa lysate digest were injected every four runs. Raw mass spectrometry data were processed using Proteome Discoverer (v2.5) with mass tolerances of 10 ppm (MS1) and 0.02 Da (MS2). Peptide spectra were searched against the UniProt protein database with the following search parameters: trypsin cleavage with up to two missed cleavages, carbamidomethylation of cysteine as a fixed modification, and methionine oxidation and N-terminal acetylation as variable modifications. Protein and peptide identification confidence was assessed via the Percolator algorithm, applying a false discovery rate (FDR) threshold of 0.01.

Label-free quantification was performed using at least two unique peptides per protein, with abundance calculations based on pairwise ratio-based comparisons, and statistical significance was determined using a t-test background-based approach. Proteins were considered significantly modulated if they exhibited an adjusted $P$-value < 0.05 and a fold change (FC) > 1.3.

**Bioinformatics analyses of proteomics data.** Enrichment analysis on significantly differentially expressed proteins was performed using PathFindeR R package. Next, enriched pathways were clustered using the SimplifyEnrichment R package, and the identified clusters were manually annotated.

### Statistical analysis

For statistical analysis, the GraphPad Prism 9.0 (v 9.0.0) was used. Data are depicted as mean with standard deviation (SD) or standard error of the mean (SEM) as indicated in the respective Figure legends. $P$-values of $P < 0.0001$ or $P < 0.001$ were considered significant, $P < 0.01$ very significant, and $P < 0.05$ significant. Data were tested with two-tailed Mann-Whitney U-test to compare unmatched groups with non-Gaussian distribution, or with two-way analysis of variance (ANOVA) followed by Turkey's multiple comparison to determine differences among multiple datasets (threshold for significance: $P \leq 0.05$). Statistical analyses are available in the data source files. The number of animals/samples used in each experiment is specified in figure legends.

**Statistical analysis of cell type proportions.** To assess differences in cell type proportions between WT and 3xTg-AD mice (only brain, only meninges, and both tissues together), we computed odds ratios (ORs) with 95% confidence intervals (CIs) for each cell type. The ORs were calculated as:

$$OR = \frac{\frac{p_{3xTg-AD}}{1-p_{3xTg-AD}}}{\frac{p_{WT}}{1-p_{WT}}} \tag{1}$$

where $p_{3xTg-AD}$ and $p_{WT}$ represent the proportions of a given cell type in 3xTg and WT mice, respectively. The standard error (SE) of the log(OR) was computed as:

$$SE = \sqrt{\frac{1}{n_{WT,yes}} + \frac{1}{n_{WT,no}} + \frac{1}{n_{3xTg-AD,yes}} + \frac{1}{n_{3xTg-AD,no}}} \tag{2}$$

where $\eta_{WT,yes}$, $\eta_{WT,no}$, $\eta_{3xxTg-AD,yes}$, $\eta_{3x}x_{Tg-AD,no}$ represents the numerosity of the given cell type (yes) or all the others (no) in WT and 3xTg-AD mice respectively. The 95% CI for the OR was obtained by exponentiating the bounds of the log(OR) ± 1.96 × SE. To evaluate the statistical significance of observed differences, we performed a permutation test (10,000 iterations). At each iteration, cell type labels were randomly reassigned between groups while maintaining the original total number of WT and 3xTg-AD cells. The empirical P-value was computed as the fraction of permuted ORs deviating from 1 as much or more than the observed OR. Multiple testing correction was applied using the False Discovery Rate (FDR) method.

### Reporting summary
Further information on research design is available in the Nature Portfolio Reporting Summary linked to this article.

## Data availability
Our datasets of scRNA-seq data are deposited online in the GEO and are publicly available under accession numbers GSE180188 and GSE180184. Human scRNA-seq datasets obtained from Xu et al.[45] and Gate et al[9]. are available under GSE181279 and GSE134578 accession codes, respectively. All data associated with this study can be found in the paper or in supplementary materials. The mass spectrometry proteomics data have been deposited to the ProteomeXchange Consortium via the PRIDE partner repository with the dataset identifier PXD064640. Raw data generated in this study are provide in the Supplementary Information and Source Data files. Source data are provided with this paper.

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

## Acknowledgements

This work was supported in part by the European Research Council (ERC) Advanced Grant no. 695714 and the ERC Proof of Concept grants no. 693606 IMPEDE and 101069397 NeutrAD; the Fondazione Italiana Sclerosi Multipla (FISM) Genova, Italy; The NextGenerationEU Program and National Recovery and Resilience Plan (PNRR), National Centers Program (National Biodiversity Future Center (CN_00000033; CUP B33C22000660001) and PNRR "Partenariati estesi" Program, grant MNESYS (PE0000006; CUP B33C22001060002) (to GC). E.Z was supported by a fellowship from Fondazione Veronesi, Milan, Italy. G.A. was supported by a fellowship from the Fondazione Italiana Sclerosi Multiple, Genoa, Italy. We thank the Center for Technological Platforms form the University of Verona for providing the genomic, proteomic and flow cytometry platforms.

## Author contributions

The study was designed by E.T., E.C.P. and G.C. E.T., G.T., M.C., N.Vi., E.Z. and M.C. performed scRNAseq and bioinformatics analysis. E.Terrabuio, E.C.P., A.B., G.F., A.C., F.M. and B.S.L. performed flow cytometry experiments. In vivo experiments and behavioral tests were performed and analyzed by E.Terrabuio, and E.C.P. Immunohistochemistry and immunofluorescence experiments on human and murine samples were

performed and analyzed by E.C.P., G.A. and C.L. B.B. provided AD samples and contributed to human data analysis. Immunofluorescence experiments on in vitro co-culture were performed by E.C.P. and E. Turano. E.Terrabuio, E.C.P, A.B, C.L. and V.D.B. conducted in vitro live-cell imaging. E.Terrabuio and B.R. analyzed in vitro live cell image data. E.Terrabuio, D.C., J.B., E.C.P. and N.Va. prepared the cells for proteomics and performed the proteomic analysis. E.T., E.C.P. and G.C. wrote the manuscript. G.C. provided the financial support.

## Competing interests

The authors declare no competing interests.
