## [Transparent Peer Review file · Nature Communications]

CD103–CD8+ T cells promote neurotoxic inflammation in Alzheimer’s disease via granzyme K–PAR-1 signaling

Corresponding Author: Professor Gabriela Constantin

Version 0:

Reviewer comments:

Reviewer #1

(Remarks to the Author)

In this manuscript, Terrabuio et al report that 1) The CD8+ Trm is dysregulated in AD, with CD103- CD8+ T cells increase and CD103+ CD8+ T decrease in the brain of 3xTg-AD mice. 2) Granzyme K (Grk) is mainly expressed in CD103- CD8+ T cells. 3) LFA-1 integrin leads the accumulation of CD103- CD8+ T cells. 4) Depletion of CD103-CD8+ T cells is protective for AD neuropathology. Overall, the research and findings are interesting for the field. There are, however, major and minor concerns that need to be addressed.

Major comments:

1. Characterization of Trm is essential for this research. Please show the Itgae (CD103) and Grk expression dot plots in Fig 1d. In Figs 1e and 1f, CD103 is widely expressed in both effectors, the CD103- and CD103+ CD8+ T cell populations.
2. Please show the distribution and quantification of both the CD103- and CD103+ CD8+ T cell populations in the trajectory analysis in Figure 2b and describe the trajectory branch genes in the supplement figure.
3. The authors showed the Grk+ CD8+ T cells were close to hippocampal neurons in 3xTg-AD mice by immunofluorescence staining of NeuN and Grk. Please include the neuronal axon or dendrite markers, as well as CD8 and Grk. The current resolution and imaging of CD8 and Grk are not convincing in Fig.2o and Fig.6.
4. The authors investigated the interaction between PAR-1 and Grk in vitro culture. Please at least show the PAR-1, CD8, and Grk staining in the brain of 3xTg-AD mice.
5. The authors globally depleted the CD8 T cells using anti-CD8a antibody, however, it will not be able to conclude that “Detrimental CD103- CD8+ Trm cells in the brain originate from the circulation and their depletion ameliorates disease in AD mice”. What we can conclude from this assay is that CD8+ T cells contribute to AD cognition in 3xTg-AD mice. To know whether it would contribute to neuropathology, 1) please show the soluble and insoluble Ab40, Ab42, tau, and tau by Elisa. 2) The Aβ DAB is not correct. Please confirm whether it is located in the extracellular or intracellular, in the form of oligomer or fibrillar?
6. The authors reported that LFA-1 integrin leads to the accumulation of CD103- CD8+ T cells. 1) Please show the total immune cell population in 3xTg-AD/Itgal-/- mice. LFA-1 also plays an important role in neutrophil recruitment. How to validate it is CD103- CD8+ Trm cells specific? 2) Please show the neuropathology in the brain of 3xTg-AD/Itgal-/-.
7. The authors reported that recombinant Grk significantly increased hyperphosphorylation of tau on AT8 and AT100. Please describe where the hyperphosphorylation tau is located, in the soma or the dendrites? Please also show either by WB or Elisa for the AT8 and AT100 changes.

Minor comments:

The citation and reference of the manuscript need to be highly improved. Some of the summaries of published work are not correct.

Reviewer #2

(Remarks to the Author)

The study investigates the role of adaptive immunity in Alzheimer's disease (AD) by focusing on CD8⁺ T cells in both human patients and a mouse model. The authors report an accumulation of activated, CD103⁻ tissue-resident memory (Trm) CD8⁺ T cells in the AD brain, characterized by high granzyme K (GrK) production. These cells, originating from the circulation and migrating into the brain via LFA-1 integrin, are implicated in neuronal dysfunction and tau hyperphosphorylation through GrK activation of protease-activated receptor-1 (PAR-1). Ablating CD103⁻CD8⁺ T cells in AD mice reportedly improved cognitive function and reduced neuropathology, suggesting a novel immune-mediated neurotoxic pathway as a potential therapeutic target.

However, the identification and characterization of CD103⁻ and CD103⁺ Trm subsets lack robust validation. The clustering data, as visualized in UMAP plots, do not convincingly distinguish these populations, raising questions about the classification criteria and gating strategies. Furthermore, the findings rely heavily on single-cell RNA sequencing without sufficient validation through orthogonal techniques like flow cytometry, which would strengthen the conclusions about cluster-specific roles in AD pathophysiology. These methodological shortcomings highlight the need for improved rigor in defining immune cell subsets and their functional roles in neurodegenerative diseases.

Major critiques

1- The observation that CD103⁻ Trm cells exhibit enhanced granzyme K expression compared to their CD103⁺ counterparts is not novel, as it has been previously reported in the context of autoimmunity, specifically in primary Sjögren's syndrome (JCI Insight, 2023 Apr 24; 8(8)).

2- I have concerns regarding the characterization of CD103⁺ versus CD103⁻ CD8⁺ T cells in Figure 1e. The UMAP plot depicting the expression of *Itgae* (the gene encoding CD103) does not clearly differentiate two distinct Trm populations based on CD103 expression. Furthermore, the data presented in Figures 1d and 1f suggest that these two Trm subsets are distinguished by low and high *Itgae* expression levels, which seems to contradict the phenotypes the authors are reporting for CD103⁻ and CD103⁺ Trm cells. Can the authors clarify these discrepancies and provide additional evidence supporting their gating and classification strategy?

3- Analyzing approximately 3,100 cells may lack the resolution needed to reliably identify and characterize specific functional Trm subsets, particularly those involved in nuanced roles such as neuroinflammation in AD. This limitation is further compounded by the absence of considerations for mouse-to-mouse variability or potential batch effects, which could affect the generalizability of the findings, especially in a dynamic disease model like AD.

To address these issues, the authors should consider expanding the dataset by integrating data from multiple animals or replicates to better capture rare Trm subsets and ensure robust statistical power. Additionally, the scRNA-seq data should be complemented with orthogonal approaches, such as flow cytometry, to fully characterize these subsets and validate their functional roles. This combined approach would strengthen the conclusions and provide a more comprehensive understanding of Trm heterogeneity in AD.

4- In Figure 1g, the authors report proportional changes in CD8⁺ T cell clusters in the brain between 3xTg-AD and WT mice. However, it is unclear whether these changes are statistically significant. The authors should perform appropriate statistical analyses to determine the significance of these differences and include the results in the figure or accompanying text. This analysis is crucial to validate the reported changes and their biological relevance.

5- The flow cytometry staining of CD103 and CD69 expression in Extended Figure 1k is confusing and lacks sufficient clarity. To improve interpretation, the authors should include a detailed gating strategy, starting from the initial population (e.g., lymphocytes) and demonstrating the sequential gating steps to isolate CD3⁺ and CD8⁺ cells. Additionally, they should provide representative flow cytometry plots for each gating step and clearly explain how the percentages of CD103⁺ and CD69⁺ cells were calculated within the CD3⁺CD8⁺ population.

6- In Figure 2k, the authors show that the majority of GZMK⁺ cells are CD103⁻, even though the frequency of CD8⁺ T cells expressing granzyme K is similar between 3xTg-AD and WT mice. However, it is surprising that in Figure 2o, CD8⁺ T cells infiltrating the brain appear to be granzyme K-negative in WT mice. This discrepancy raises questions about the consistency of granzyme K expression across the datasets.

7- Additionally, it would be critical to demonstrate the comparative granzyme K expression in CD103⁻ versus CD103⁺ CD8⁺ T cells specifically in the brain of AD mice. This analysis would help clarify whether the reported differences in granzyme K expression are tied to the CD103⁺/CD103⁻ status of Trm cells and provide greater insight into their functional roles in the AD brain.

8- The extent to which findings from the 3xTg-AD mouse model translate to human AD remains unclear. Recent studies indicate that Trm CD8⁺CD103⁺ cells are increased in the brains of AD mice (APP-PS1) compared to WT mice (Altendorfer, J. Immunol., 2022). Notably, in human AD, a more recent study demonstrated an increase in CD8⁺CD103⁺ cells in the cerebrospinal fluid (CSF) of AD patients compared to healthy controls (Kimura, Neurol. Neuroimmunol. Neuroinflamm., 2023). These findings appear to contradict the results presented in the current study. How do the authors account for these discrepancies?

9- While GrK is implicated as a neurotoxic mediator, the exact intracellular and downstream signaling pathways triggered by GrK-PAR-1 activation are not detailed. For instance, does GrK lead to oxidative stress, synaptic pruning, or other specific neurotoxic effects?

Minor

1- In the introduction line 78, Temra cells should be defined as C45RA+CCR7-

Version 1:

Reviewer comments:

Reviewer #1

(Remarks to the Author)

The authors have addressed the majority of the questions raised during the revision.

Reviewer #2

(Remarks to the Author)

The authors have satisfactorily addressed the major critiques. Regarding the characterization of CD103⁺ versus CD103⁻ CD8⁺ T cells (Figure 1e), the authors have acknowledged the ambiguity in Itgae expression and have now provided additional analyses to better support the distinction between CD103⁺ and CD103⁻ subsets although the distinction remains somewhat gradated rather than binary.

Point by point answers to the reviewers' comments

Reviewer #1 (Remarks to the Author):

In this manuscript, Terrabuio et al report that 1) The CD8⁺ Trm is dysregulated in AD, with CD103⁻ CD8⁺ T cells increase and CD103⁺ CD8⁺ T decrease in the brain of 3xTg-AD mice. 2) Granzyme K (Grk) is mainly expressed in CD103⁻ CD8⁺ T cells. 3) LFA-1 integrin leads the accumulation of CD103⁻ CD8⁺ T cells. 4) Depletion of CD103⁻ CD8⁺ T cells is protective for AD neuropathology. Overall, the research and findings are interesting for the field. There are, however, major and minor concerns that need to be addressed.

Major comments:

1. Characterization of Trm is essential for this research. Please show the Itgae (CD103) and Grk expression dot plots in Fig 1d. In Figs 1e and 1f, CD103 is widely expressed in both effectors, the CD103⁻ and CD103⁺ CD8⁺ T cell populations.

As requested, we have now added *Gzmk* gene expression to the dot plot in Fig. 1d (*Itgae* gene expression was already shown). To clarify the differences in gene expression in Fig. 1f, we now show the violin plots as “extended” versions and have added the median gene expression values (white dashed lines). Fig 1f now clearly shows that the CD103⁺ Trm cluster is characterized by a high median level of *Itgae* gene expression (median = 3.13, although some cells express lower levels) compared to CD103⁻ Trm cells (median = 1.75) and effector cells (median = 1.81), despite the presence of some *Itgae*-expressing cells in both these latter clusters. These results, although showing some heterogeneity, suggest that *Itgae* gene expression in the CD103⁺ Trm cluster is more homogeneous than in the CD103⁻ Trm and effector populations.

In our study, *Itgae* is used as one of several biomarker genes for the CD103⁺ Trm cluster and is evaluated using PartekFlow software considering both the averaged gene expression value and the number of cells expressing the gene, as reported in Supplementary Information 1. However, the residual *Itgae* gene expression we observed in the CD103⁻ Trm population is in line with previous studies of CD103⁺ and CD103⁻ Trm cells sorted from the mouse brain for transcriptomic studies, revealing that some CD103⁻ Trm CD8⁺ cells have residual *Itgae* gene expression (Wakim, L.M. *et al.* 2012). More recent reports have confirmed the heterogeneity of *Itgae* gene expression in the CD103⁺ and CD103⁻ populations. For example, the IL7R⁺CD103⁺ cluster of CD8⁺ T cells in the brain of aged mice has non-homogeneous *Itgae* expression, with several cells not expressing this gene (Groh, J. *et al.* 2021). Similarly, *Itgae* expression is neither homogeneous nor exclusively segregated in the CD103⁺ Trm CD8⁺ T cell population in the human gut lamina propria, but can be also present in CD103⁻ cells (FitzPatrick, M.E.B., *et al.* 2020). Furthermore, *Itgae* expression is also present in the CD103⁻ Trm CD8⁺ T cell population in the gut of patients with Chron's disease, albeit at lower levels than observed in CD103⁺ Trm CD8⁺ T cells (Bottois, H. *et al.* 2020). Overall, the heterogeneity of *Itgae* expression suggests that this gene cannot be considered as a strict marker for the identification of CD103⁺ Trm CD8 cells, but must be evaluated together with other marker genes to distinguish between CD103⁺ and CD103⁻ Trm CD8⁺ T cells (Carbone, F.R. 2015; Herndler-Bransdetter, D. *et al.* 2018; FitzPatrick M.E.B *et al.*, 2012; Crowl JT *al.*, 2022). Indeed,

the UMAP plots in Fig. 1e show that *Gstp3* and *Foxo1*, which have also been proposed as CD103⁺ Trm phenotypic markers (Kumar B.V. et al., 2017; Milner J.J. et al., 2018; FitzPartick M.E.B et al., 2021), are preferentially expressed in cells of the CD103⁺ Trm cluster. In addition, CD103⁺ cells express low levels of *Eomes*, which is more strongly expressed in the CD103⁻ Trm population, further strengthening our Trm cell classification. The UMAP plots in Fig. 1e also show that although the *Itgae* gene is expressed in a few cells of the effector cluster, it is characterized by the almost exclusive expression of *Gzma*, *Slpr5* and *Cx3cr1*, and is therefore clearly segregated from the Trm CD103⁺ and CD103⁻ CD8⁺ T cell populations. In Fig. 1f, we also observed a high median expression level of the *Itgae* gene in the Tcm cluster, although Fig. 1e shows that only a few Tcm cells express this gene. Moreover, the Tcm cluster is also clearly segregated from the Trm CD103⁺ and CD103⁻ clusters of CD8⁺ T cells, and is characterized by the almost exclusive expression of *Sell*, *Ccr7* and *Nsg2*, which classically characterize the Tcm phenotype (Dean J.W. et al., 2023; Buquicchio F.A. et al., 2024). Therefore, despite some heterogeneity in *Itgae* gene expression, our data show well-segregated CD8⁺ T cell clusters based on the analysis of scRNAseq data.

To validate the scRNAseq data and better characterize the CD8⁺ T cell population in our AD model, and to fully address the point raised by the reviewer, we performed additional flow cytometry experiments. In line with previous studies (Smolders J., et al., 2018), our new data show that both the CD103⁺ and CD103⁻ Trm cell populations were positive for CD69. Notably, CD103⁺ cells had lower intracellular EOMES protein levels compared to CD103⁻ CD8 Trm cells, confirming the scRNAseq data. As expected, effector cells were CD69⁻GrA⁺GrB⁺, whereas Tcm cells were CD69⁻CD62L⁺. Very few Trm^{PROL} cells were found in the scRNAseq dataset and were not characterized by flow cytometry. Notably, in line with other data already shown at submission in Fig. 2n our new data confirm that GrK expression is significantly higher in CD103⁻ than CD103⁺ cells. These new results supporting the scRNAseq data are shown in Fig. 1g, h, whereas the gating strategy is included in Extended Data Figs 5f-h and the methodology is described on p47-49.

2. Please show the distribution and quantification of both the CD103⁻ and CD103⁺ CD8⁺ T cell populations in the trajectory analysis in Figure 2b and describe the trajectory branch genes in the supplement figure.

We now provide the distribution and quantification of the CD103⁻ and CD103⁺ Trm CD8⁺ T cell populations from Fig. 2b along the trajectory plot in Extended Data Fig. 1f, g. Also, we describe the trajectory branch genes in the legend of Extended Data Fig. 1i, as suggested by the reviewer. We would like to emphasize that unbiased cell fate trajectory analysis showed only slight differences in the meninges whereas there were clear differences in the brain (Fig. 2c; Extended Data Fig. 1h-l), where Arm A of the trajectory plot was populated by a lower proportion of CD103⁺CD8⁺ Trm cells in 3xTg-AD mice compared to WT controls accompanied by a higher percentage of CD103⁻CD8⁺ Trm cells in Arm B (Fig. 2c; Extended Data Fig. 1h-l). These data are supported by flow cytometry results (Fig 2d, e; Extended Data Fig. 1n-o), further highlighting the brain as a fundamental point of CD8⁺ T cell dysregulation in AD-like mice.

3. The authors showed the GrK⁺ CD8⁺ T cells were close to hippocampal neurons in 3xTg-AD mice by immunofluorescence staining of NeuN and Grk. Please include the neuronal axon or

dendrite markers, as well as CD8 and Grk. The current resolution and imaging of CD8 and Grk are not convincing in Fig. 2o and Fig.6.

As requested, for Fig. 2o (now Fig. 2p), we replaced NeuN with a pan-neuronal marker (Milli-Mark MAB2300), which stains the architecture of the whole neuron, including the axon and dendrites. To improve the resolution, we acquired z-stacked images using the Apotome 2 module (Zeiss), which significantly reduces noise during acquisition. The z-stacked images were then post-processed with Imaris software, and a Gaussian filter was applied to remove even more noise. In the new images, the signal of the pan-neuronal marker is white to enhance the contrast and better visualize the neuronal projections, while GrK and CD8 are green and red, respectively. Our new images confirmed the presence of GrK⁺CD8⁺ T cells near the hippocampal neurons of 3xTg-AD mice, whereas CD8⁺ T cells expressing low GrK levels were mainly detected in the hippocampus of sex- and age-matched WT controls. To improve Fig. 6i, we have increased the resolution by applying a Gaussian filter using Imaris software, thus reducing the noise. Although the image was not acquired in z-stacks, we believe that the low image resolution noted by the reviewer was caused by embedding of the images in the final PDF file generated by the system. For this reason, we now provide each figure as a separate file.

4. The authors investigated the interaction between PAR-1 and Grk in vitro culture. Please at least show the PAR-1, CD8, and Grk staining in the brain of 3xTg-AD mice.

We provide new immunofluorescence images of brain slices from 3xTg-AD mice showing that GrK⁺ (green) CD8⁺ (red) T cells are located near neurons (white) expressing PAR-1 (orange). Neurons were stained using a pan-neuronal marker to visualize the whole cellular architecture, as described above. The new data, shown in Extended Data Fig. 2i, suggest that GrK⁺CD8⁺ T cells are positioned near axons and dendrites in the brains of AD mice, in line with previous studies showing CD8⁺ T cells near neuronal processes in the hippocampus of AD patients and APP/PS1 mice as well in the white matter of aged mice (Gate, D. et al. 2020; Groh et al., 2021). As expected, brain CD8⁺ T cells were also positive for PAR-1 given that the receptor plays a role in CD8⁺ T cell activation and degranulation (Chen, H. et al. 2021).

5. The authors globally depleted the CD8 T cells using anti-CD8a antibody, however, it will not be able to conclude that “Detrimental CD103⁻ CD8⁺ Trm cells in the brain originate from the circulation and their depletion ameliorates disease in AD mice”. What we can conclude from this assay is that CD8⁺ T cells contribute to AD cognition in 3xTg-AD mice. To know whether it would contribute to neuropathology, 1) please show the soluble and insoluble Ab40, Ab42, tau, and tau by Elisa. 2) The A β DAB is not correct. Please confirm whether it is located in the extracellular or intracellular, in the form of oligomer or fibrillar?

Previous reports suggest that the CD103⁻ enriched Trm CD8⁺ T cell population is derived from circulating exKLRG1 effector CD8⁺ T cells retaining a high cytotoxic capacity in mice (Herndler-Brandstetter, D., et al. 2018). Although we do not directly show that CD8⁺ T cells migrate from the blood into the AD brain and contribute to the formation of the CD103⁻CD8⁺ Trm population, our data strongly suggest a peripheral origin for these cells. Indeed, it was previously shown that the treatment of mice with an anti-CD8 antibody depletes circulating CD8⁺ T cells, leaving the brain

Trm compartment unaltered (Steinbach, K. *et al.* 2016; Urban, S.L. *et al.* 2020). In this context, the significant loss of CD103⁻ Trm CD8⁺ T cells in the brain of CD8-depleted 3xTg-AD mice compared to isotype controls (Fig. 4a) clearly indicates that replenishment from the circulation is limited. In contrast, the CD103⁺ Trm CD8⁺ population was unchanged after depletion in 3xTg-AD mice, indicating that this population does not rely on circulating CD8⁺ T cells (Fig. 4a). Notably, the significant reduction of the CD103⁻CD8⁺ Trm population in the brains of CD8-depleted 3xTg-AD mice was paralleled by a clear improvement of cognitive behavior and neuropathological alterations (Fig. 4 b-m), strongly suggesting that CD103⁻ Trm CD8⁺ T cells are replenished from the blood and promote the induction of cognitive deficits in AD mice.

As requested, we conducted ELISA experiments on soluble and insoluble fractions from brain homogenates (Illouz, T. *et al.* 2017), focusing on soluble and insoluble ptau (pT231), total tau, Aβ1–40 and Aβ1–42. These new data (Results p10 and Fig. 4 j-l) show a significant loss of soluble and insoluble pTau, Aβ1–40 and Aβ1–42 in the brains of 3xTg-AD mice depleted of peripheral CD8⁺ T cells compared to 3xTg-AD isotype controls, confirming our neuropathological studies (Fig. 4g-i). We also performed dot blot experiments on brain homogenates and detected a significant loss of insoluble oligomeric (A11 antibody) and fibrillar (OC antibody) forms of Aβ in 3xTg-AD mice depleted of CD8⁺ T cells compared to isotype controls (Fig. 4m). As expected, we detected very low amounts of soluble_oligomeric and fibrillar forms of Aβ (Extended Data Fig. 3i, j), and observed no significant difference between 3xTg-AD mice depleted of CD8⁺ T cells and isotype controls, as previously shown (Oddo S. *et al.* 2005). These authors detected a low amount of high-molecular-weight oligomers (A11 signal) and fibrils (OC signal) in the soluble fraction of 9-month-old 3xTg-AD mice.

We also performed immunofluorescence staining on brain slices with A11 (high-molecular-weight Aβ oligomers), OC (Aβ fibrils), and 6E10 (all forms of Aβ) antibodies to further confirm our immunohistochemical staining of Aβ (Fig. 4g) and to study the localization of Aβ oligomers and fibrils in the hippocampus of 3xTg-AD mice. Our data showed a clear co-localization of 6E10 positivity with the A11 and OC signals at the intra-neuronal level in 9-month-old AD mice in line with previous studies (Pensalfini A., *et al.*, 2014) (Fig. 4n). Overall, our neuropathology data is now stronger, and we would like to thank the reviewer for this comment, which improved our work.

6. The authors reported that LFA-1 integrin leads to the accumulation of CD103- CD8+ T cells. 1) Please show the total immune cell population in 3xTg-AD/Itgal-/- mice. LFA-1 also plays an important role in neutrophil recruitment. How to validate it is CD103- CD8+ Trm cells specific? 2) Please show the neuropathology in the brain of 3xTg-AD/Itgal-/-.

LFA-1 integrin contributes to a broad spectrum of molecular mechanisms underpinning both innate and adaptive immune responses, including neutrophil recruitment in the brains of AD mice, as we previously demonstrated (Zenaro, E. *et al.* 2015). As shown in Fig. 5c, our data clearly indicate that LFA-1 integrin controls the accumulation of CD103⁻ CD8⁺ cells in the brains of 3xTg-AD mice because AD mice deficient in LFA-1 (3xTg-AD/*Itgal*^{-/-}) have fewer of these cells, suggesting a crucial role for LFA-1 in the accumulation of CD103⁻CD8⁺ Trm cells in the brain and strengthening our proposed peripheral origin of brain CD103⁻ Trm CD8⁺ T cells in 3xTg-AD mice. As suggested

by the reviewer, we updated Fig. 5b with the total immune CD45⁺ cell population detected in the brains of 3xTg-AD/*Itgal*^{-/-} mice compared to sex- and age-matched 3xTg-AD controls.

To fully address reviewer's comments, we have also added new data to Fig. 5d-f showing a significant reduction of the A β load and tau hyperphosphorylation (AT180) in the brains of 3xTg-AD/*Itgal*^{-/-} mice compared to sex- and age-matched 3xTg-AD controls, but no significant difference in the levels of total tau, further supporting a detrimental role for pathogenic CD8⁺ T cells. The new data are described in the updated Results section (p10).

7. The authors reported that recombinant Grk significantly increased hyperphosphorylation of tau on AT8 and AT100. Please describe where the hyperphosphorylation tau is located, in the soma or the dendrites? Please also show either by WB or Elisa for the AT8 and AT100 changes.

Our immunofluorescence experiments on differentiated human SH-SY5Y cells (Fig. 7d, e; Extended Data Fig. 4f, g) indicated that the AT100 signal was localized in both the soma and dendrites of neurons, whereas the AT8 signal was mainly located in the soma. Co-staining with MAP2 was used to visualize the whole neuronal structure (Extended Data Fig. 4f, g). These results are now better described in the main text (Results, p12).

As requested, we performed new ELISA experiments confirming that recombinant GrK induces phosphorylation on serine, but not on threonine, residues of tau protein in these cells. Indeed, these new data reveal a significant increase of phosphorylation on tau protein residues pS199 (AT8) and pS396, but not on pT231 (AT180). The new data are described in the Results (p12) and are shown in Fig. 7g.

Minor comments:

The citation and reference of the manuscript need to be highly improved. Some of the summaries of published work are not correct.

We have checked the references and made the necessary changes in the citations. We thank the reviewer for bringing this to our attention.

Reviewer #2 (Remarks to the Author):

The study investigates the role of adaptive immunity in Alzheimer's disease (AD) by focusing on CD8⁺ T cells in both human patients and a mouse model. The authors report an accumulation of activated, CD103⁻ tissue-resident memory (Trm) CD8⁺ T cells in the AD brain, characterized by high granzyme K (GrK) production. These cells, originating from the circulation and migrating into the brain via LFA-1 integrin, are implicated in neuronal dysfunction and tau hyperphosphorylation through GrK activation of protease-activated receptor-1 (PAR-1). Ablating CD103⁻CD8⁺ T cells in AD mice reportedly improved cognitive function and reduced neuropathology, suggesting a novel immune-mediated neurotoxic pathway as a potential therapeutic target.

However, the identification and characterization of CD103⁻ and CD103⁺ Trm subsets lack robust validation. The clustering data, as visualized in UMAP plots, do not convincingly distinguish these

populations, raising questions about the classification criteria and gating strategies. Furthermore, the findings rely heavily on single-cell RNA sequencing without sufficient validation through orthogonal techniques like flow cytometry, which would strengthen the conclusions about cluster-specific roles in AD pathophysiology. These methodological shortcomings highlight the need for improved rigor in defining immune cell subsets and their functional roles in neurodegenerative diseases.

Major critiques

1- The observation that CD103⁻ Trm cells exhibit enhanced granzyme K expression compared to their CD103⁺ counterparts is not novel, as it has been previously reported in the context of autoimmunity, specifically in primary Sjögren's syndrome (JCI Insight, 2023 Apr 24; 8(8)).

We are aware that an increased number of GrK⁺ Trm CD8⁺ T cells has been reported in the context of certain autoimmune disorders, including Sjögren's syndrome (Xu, T. et al, 2023) and multiple sclerosis (Koetzier, S.C. et al, 2022), as also stated by the reviewer. In addition, a study performed on a heterogeneous group of brain samples showed that more GrK accumulated in brain CD103⁻CD69⁺ T cells compared to CD103⁺CD69⁺ T cells, but most brain donors had no neurological disease or were patients with multiple sclerosis, Parkinson's diseases or bipolar disorder, and no data correlated with AD or could be attributed to a specific brain disorder (Smolders J. et al. 2018). Given the above context, we studied meningeal and brain CD8⁺ T cell compartments at the single-cell level and demonstrated that the main alterations occur in the brains of mice with both amyloid and tau pathologies during early disease stages. We clearly showed for the first time an increase in the population of activated CD103⁻CD8⁺ T cells producing GrK (but not GrA, GrB or perforin) in 3xTg-AD mice, and this subset was also more abundant in human AD patients, highlighting the clinical relevance of our preclinical data. We demonstrate that CD103⁻CD8⁺ T cells have a detrimental role in disease pathogenesis in AD mice and show for the first time they have a circulation origin and migrate into the brain using LFA-1 integrin.

It is largely unclear how CD8⁺ T cells contribute to brain dysfunction, but we provide the novel contribution that CD103⁻CD8⁺ T cells producing GrK induce functional alterations in mouse primary neurons. Moreover, we show that active purified GrK directly induces neuronal dysfunction in both mouse and human neuronal cells. Supporting the clinical relevance of these new data, we found that GrK⁺CD103⁻CD8⁺ T cells accumulate in the brains of AD patients and are positioned near neuronal cells.

To further demonstrate the mechanistic relevance of our data we found that GrK induces tau hyperphosphorylation and functional neuronal alterations by interacting with the neuronal PAR-1 receptor, revealing a previously unknown immune-mediated neurotoxic axis in AD. Furthermore, our new proteomics data (now shown in Fig. 7h-i) offer mechanistic information clearly showing that Grk induces key molecular pathways involved in the pathogenesis of AD. PAR-1 was expressed at higher levels in the brains of 3xTg-AD mice and inhibiting the dysregulated GrK–PAR-1 axis reduced neurotoxicity, highlighting the relevance of this new molecular mechanism in AD-related chronic neuroinflammation and neurodegeneration. Importantly, given that PAR-1-targeting drugs such as vorapaxar are already used in clinic to block platelet aggregation (Zang C. et al, 2012), our results suggest that the manipulation of GrK–PAR-1 interactions may inhibit AD

pathogenesis, offering a novel therapeutic approach targeting immune mechanisms in neurodegenerative diseases.

Overall, we believe that our data present several original elements showing that GrK is the basis of a new immune mechanism promoting brain neurotoxicity with a key role in the mediation of CD103⁻CD8⁺ Trm cell-dependent neuronal alterations in AD.

*2- I have concerns regarding the characterization of CD103⁺ versus CD103⁻ CD8⁺ T cells in Figure 1e. The UMAP plot depicting the expression of *Itgae* (the gene encoding CD103) does not clearly differentiate two distinct Trm populations based on CD103 expression. Furthermore, the data presented in Figures 1d and 1f suggest that these two Trm subsets are distinguished by low and high *Itgae* expression levels, which seems to contradict the phenotypes the authors are reporting for CD103⁻ and CD103⁺ Trm cells. Can the authors clarify these discrepancies and provide additional evidence supporting their gating and classification strategy?*

We agree with the reviewer that the difference between the CD103⁺ and CD103⁻ CD8⁺ T cell populations is not defined by on/off *Itgae* expression, and we would like to raise several considerations and report the results of new experiments to support our claims.

First, to clarify differences in *Itgae* expression, we now show the violin plots as “extended” versions and have added the median gene expression levels (white dashed lines) in Fig. 1f. These data clearly show that the CD103⁺ Trm cluster is characterized by a high median level of *Itgae* gene expression (median = 3.13, although some cells express lower levels) compared to CD103⁻ Trm cells (median = 1.75) and effector cells (median = 1.81), despite the presence of some *Itgae*-expressing cells in both these latter clusters. These results (together with Fig. 1e), although showing some heterogeneity, suggest that *Itgae* gene expression in the CD103⁺ Trm cluster is more homogeneous than in the CD103⁻ Trm and effector populations.

Second, in our study, *Itgae* gene is one of the biomarker genes of the CD103⁺ Trm cluster, calculated by the PartekFlow software considering both the averaged gene expression and the number of cells expressing the gene, as reported in the Supplementary Information 1. However, the residual *Itgae* gene expression we observed in the CD103⁻ Trm population is in line with previous studies of CD103⁺ and CD103⁻ Trm cells sorted from the mouse brain by RNAseq, revealing that some CD103⁻ Trm CD8⁺ cells have residual *Itgae* gene expression (Wakim, L.M. *et al.* 2012). These results confirm our observations and suggest that gene expression and protein abundance evaluated by transcriptomic and proteomic approaches, respectively, may not fully overlap. Indeed, more recent reports have confirmed the heterogeneity of *Itgae* gene expression in the CD103⁺ and CD103⁻ populations. For example, the IL7R⁺CD103⁺ cluster of CD8⁺ T cells in the brain of aged mice has non-homogeneous *Itgae* expression, with several cells not expressing this gene (Groh, J. *et al.* 2021). Similarly, *Itgae* expression is neither homogeneous nor exclusively segregated in the CD103⁺ Trm CD8⁺ T cell population in the human gut lamina propria, but can also be present in CD103⁻ cells (FitzPatrick, M.E.B., *et al.* 2020). Furthermore, *Itgae* expression is also present in the CD103⁻ Trm CD8⁺ T cell population in the gut of patients with Chron’s disease, albeit at lower levels than observed in CD103⁺ Trm CD8⁺ T cells (Bottois, H. *et al.* 2020). Considering these data and the previously reported *Itgae* expression levels (Wakim, L.M. *et al.* 2012), we feel that we can keep the classical CD103⁻ and CD103⁺ annotation of these subsets.

Third, the heterogeneity of *Itgae* expression suggests this gene cannot be considered as a strict marker for the identification of CD103⁺ Trm CD8 cells, but must be evaluated together with

other marker genes to distinguish between CD103⁺ and CD103⁻ Trm CD8⁺ T cells (Carbone, F.R. 2015; Herndler-Bransdetter, D. et al, 2018; FitzPatrick M.E.B et al., 2012; Crowl JT al., 2022). Indeed, the UMAP plots in Fig. 1e show that *Gstp3* and *Foxo1*, which have also been proposed as CD103⁺ Trm phenotypic markers (Kumar B.V. et al., 2017; Milner J.J. et al., 2018; FitzPartick M.E.B et al., 2021), are preferentially expressed in cells of the CD103⁺ Trm cluster. In addition, CD103⁺ cells express low levels of *Eomes*, which is more strongly expressed in the CD103⁻ Trm population, further strengthening our Trm cell classification. The UMAP plots in Fig. 1e also show that although the *Itgae* gene is expressed in a few cells of the effector cluster, it is characterized by the almost exclusive expression of *Gzma*, *Slpr5* and *Cx3cr1*, and is therefore clearly segregated from the Trm CD103⁺ and CD103⁻ CD8⁺ T cell populations. In Fig. 1f, we also observed a high median expression level of the *Itgae* gene in the Tcm cluster, although Fig. 1e shows that only a few Tcm cells express this gene. Moreover, the Tcm cluster is also clearly segregated from the Trm CD103⁺ and CD103⁻ clusters of CD8⁺ T cells, and is characterized by the almost exclusive expression of *Sell*, *Ccr7* and *Nsg2*, which classically characterize the Tcm phenotype (Dean J.W., et al., 2023; Buquicchio F.A. et al., 2024). Therefore, despite some heterogeneity in *Itgae* gene expression, our data show well-segregated CD8⁺ T cell clusters based on the analysis of scRNAseq data.

Finally, to validate the scRNAseq data and better characterize the CD8⁺ T cell population in our AD model, and to fully address the point raised by the reviewer, we performed additional flow cytometry experiments. In line with previous studies (Smolders J. et al., 2018), our new data show that both the CD103⁺ and CD103⁻ Trm cell populations were positive for CD69. Notably, CD103⁺ cells had lower intracellular EOMES protein levels compared to CD103⁻ CD8⁺ Trm cells, confirming the scRNAseq data. As expected, effector cells were CD69⁻GrA⁺GrB⁺, whereas Tcm cells were CD69⁻CD62L⁺. Very few Trm^{PROL} cells were found in the scRNAseq dataset and were not characterized by flow cytometry. Notably, in line with other data already shown at submission in Fig. 2n, our new data confirm that GrK expression is significantly higher in CD103⁻ than CD103⁺ cells. These new results supporting the scRNAseq data are shown in Fig. 1g, h, whereas the gating strategy is included in Extended Data Figs 5f-h and the methodology is described on p47-49.

3- Analyzing approximately 3,100 cells may lack the resolution needed to reliably identify and characterize specific functional Trm subsets, particularly those involved in nuanced roles such as neuroinflammation in AD. This limitation is further compounded by the absence of considerations for mouse-to-mouse variability or potential batch effects, which could affect the generalizability of the findings, especially in a dynamic disease model like AD. To address these issues, the authors should consider expanding the dataset by integrating data from multiple animals or replicates to better capture rare Trm subsets and ensure robust statistical power.

Additionally, the scRNA-seq data should be complemented with orthogonal approaches, such as flow cytometry, to fully characterize these subsets and validate their functional roles. This combined approach would strengthen the conclusions and provide a more comprehensive understanding of Trm heterogeneity in AD.

Our scRNAseq data showed that, among the 3098 CD8⁺ T cells detected in the brain and meninges of WT (n = 8) and 3xTg-AD (n = 8) mice, the majority (n = 2606) belonged to the Trm cluster (CD103⁺, CD103⁻ and Trm^{PROL}). This number of cells (n = 2606) allowed the unambiguous

detection of three subclusters: CD103⁺ Trm, CD103⁻ Trm and Trm^{PROL}. Each subcluster was characterized by the significant and relevant overexpression of multiple biomarker genes (Supplementary Information 1), which defined, for each cell population, a signature profile comparable to that described in the literature (Herndler-Brandstetter D. et al., 2018; Fung H.Y. et al., 2022; Anadon C.M. et al., 2022; Lay W. et al., 2022; Kumar B.V. et al., 2017; Milner J.J. et al., 2028; FitzPatrick M.E.B. et al., 2021). Globally, this confirms that 3098 cells provided sufficient resolution to identify and phenotypically characterize the Trm CD8⁺ T cell subsets in our dataset.

Mouse-to-mouse variability was already considered in our dataset, which is why we pooled cells from eight WT and eight 3xTg-AD mice of the same sex and age, as described in Fig. 1 legend and the Materials and Methods (p45). This ensured that the results achieved sufficient statistical power. To improve clarity, we have now included this information in the main text (p5). To validate the scRNAseq data and better characterize the CD8⁺ T cell population in our AD model, we performed additional flow cytometry experiments as described above. Our new results support the scRNAseq data (Fig. 1g, h). The gating strategy is shown in Extended Data Figs 5f-h and the methodology is described on p47-49.

4- In Figure 1g, the authors report proportional changes in CD8⁺ T cell clusters in the brain between 3xTg-AD and WT mice. However, it is unclear whether these changes are statistically significant. The authors should perform appropriate statistical analyses to determine the significance of these differences and include the results in the figure or accompanying text. This analysis is crucial to validate the reported changes and their biological relevance.

To better support the data shown in Fig. 1g (now Fig. 1j), we have included statistical analysis to evaluate the probability that each sub-population of CD8⁺ T cells would be observed under certain conditions compared to others. The statistical method is now described in the Materials and Methods section (p58, 59), and confirms that CD103⁻ Trm CD8⁺ T cells are statistically more likely to be observed in the brains of 3xTg-AD mice than in WT controls (odds ratio = 3.30; $P = 0$), whereas CD103⁺ Trm CD8⁺ cells are statistically less likely to be observed in the brains of 3xTg-AD mice than in WT controls (odds ratio = 0.29; $P = 0.0038$). This is in line with our flow cytometry data (Fig. 2 d, e). Notably, no significant difference was observed when comparing the probability to observe Teff, Tcm and Trm^{PROL} cells in the brains of 3xTg-AD mice compared to WT controls. We have described these new results in the main text (p6) and provide the analysis as data source Fig. 1.

5- The flow cytometry staining of CD103 and CD69 expression in Extended Figure 1k is confusing and lacks sufficient clarity. To improve interpretation, the authors should include a detailed gating strategy, starting from the initial population (e.g., lymphocytes) and demonstrating the sequential gating steps to isolate CD3⁺ and CD8⁺ cells. Additionally, they should provide representative flow cytometry plots for each gating step and clearly explain how the percentages of CD103⁺ and CD69⁺ cells were calculated within the CD3⁺CD8⁺ population.

The representative pseudocolor plot initially shown in Extended Data Fig. 1k (now Extended Data Fig. 1m) was intended to show that 99.2% of CD8⁺ T cells in the brain were CD69⁺ (red gate), emphasizing the need to study dysregulations of the Trm compartment in the AD brain. All gating strategies for our flow cytometry studies are provided in Extended Data Fig. 5, together with the

percentages for all gates. Particularly, the gating strategy and how we calculated the percentages of CD103⁻CD69⁺ and CD103⁺CD69⁺ Trm CD8⁺ T cell within the CD8⁺ population are described in both the Material and Methods section (p47-49) and in the legend of Extended Data Fig. 5a, b, f-h.

6- In Figure 2k, the authors show that the majority of GZMK⁺ cells are CD103⁻, even though the frequency of CD8⁺ T cells expressing granzyme K is similar between 3xTg-AD and WT mice. However, it is surprising that in Figure 2o, CD8⁺ T cells infiltrating the brain appear to be granzyme K-negative in WT mice. This discrepancy raises questions about the consistency of granzyme K expression across the datasets.

Our results presented in Fig. 2k show that the frequency of brain GrK⁺CD103⁻ Trm CD8⁺ T cells is 59.11% in WT and 85.90% in 3xTg-AD brains. Although the abundance of GrK⁺ Trm CD8⁺ T cells is not much higher in 3xTg-AD mice than in WT animals, as also noted by the reviewer, the difference resides in the level of GrK expression between these two experimental conditions. Indeed, CD103⁺ Trm CD8⁺ T cells, which express GrK protein at lower levels than the CD103⁻ Trm population (Fig. 1g, h containing new data, Fig. 2o), were present at a frequency of 40.51% in the brains of WT mice but only 7.31% in the brains of 3xTg-AD mice (Fig. 2k). Accordingly, the detection of CD8⁺ T cells expressing lower levels of GrK was almost six times more likely in WT brains compared to those of 3xTg-AD mice. This is in line with our immunofluorescence data, which showed the presence of GrK⁺ CD8⁺ T cells expressing lower levels of GrK in the brains of WT mice (Fig. 2p). We agree with the reviewer that the immunofluorescence images originally shown in Fig. 2o did not correlate well with the flow cytometry data. Therefore, we now provide a representative image showing the presence of a GrK⁺CD8⁺ T cell in the hippocampus of a WT mouse (Extended Data Fig. 2j), confirming the results obtained by flow cytometry. In the revised manuscript, we also replaced the images in Fig. 2o (now Fig. 2p) with higher-resolution images in which neurons are stained with a pan-neuronal marker rather than NeuN to better visualize the neuronal architecture (as requested by Reviewer #1). This new figure shows that CD8⁺ T cells may express low levels of GrK in the hippocampus of WT mice. We would also like to emphasize that our flow cytometry and immunofluorescence staining data may not completely overlap. Although both are fluorescence-based techniques, they differ in terms of sample preparation, data output and analysis. Most importantly, flow cytometry is global, representing all cells isolated from all brain areas, whereas immunofluorescence provides only local analysis, which depends on tissue topology and the target area. Because CD8⁺ T cells are not uniformly present in all brain slices, it is more difficult to generate matching results between immunofluorescence and flow cytometry. Despite these differences, we observed the presence of brain GrK⁺CD8⁺ T cells using both approaches in WT and 3xTg-AD mice. Our new data thus confirm the consistency between datasets. We would like to thank the reviewer for this constructive comment which improved the quality of our data.

7- Additionally, it would be critical to demonstrate the comparative granzyme K expression in CD103⁻ versus CD103⁺ CD8⁺ T cells specifically in the brain of AD mice. This analysis would help clarify whether the reported differences in granzyme K expression are tied to the CD103⁺/CD103⁻ status of Trm cells and provide greater insight into their functional roles in the AD brain.

As suggested by the reviewer, we have added violin plots to Fig. 2o comparing the expression of GrK between CD103⁻ and CD103⁺ Trm CD8⁺ T cells in the brains of 3xTg-AD mice. These violin

plots clearly show that GrK expression levels are higher in the CD103⁻ subset (median MFI = 9739) than the CD103⁺ subset (median MFI = 7619) of Trm CD8⁺ T cells in the brains of 3xTg-AD mice, in line with representative histograms (Fig. 2n). The data are mentioned in the updated Results section (p8).

8- The extent to which findings from the 3xTg-AD mouse model translate to human AD remains unclear. Recent studies indicate that Trm CD8⁺CD103⁺ cells are increased in the brains of AD mice (APP-PS1) compared to WT mice (Altendorfer, J. Immunol., 2022). Notably, in human AD, a more recent study demonstrated an increase in CD8⁺CD103⁺ cells in the cerebrospinal fluid (CSF) of AD patients compared to healthy controls (Kimura, Neurol. Neuroimmunol. Neuroinflamm., 2023). These findings appear to contradict the results presented in the current study. How do the authors account for these discrepancies?

In the Altendorfer study (Altendorfer, B. *et al.* 2022), CD8⁺ T cells were isolated from the total mouse brain without removing the meninges and choroid plexus, most likely profiling a mix of meningeal and brain subpopulations and not (as in our study) solely the parenchymal brain CD8⁺ T cells. Given our data clearly showing that the brain and meninges are characterized by different CD8⁺ T cell populations in AD, mixing the brain and meninges is a limitation, which was also acknowledged by the authors themselves in their discussion. Altendorfer and colleagues also studied CD8⁺ T cell alterations in a mouse model of amyloidosis without tau pathology, whereas A β and tau pathologies were shown to have a synergistic effect in AD and the diagnosis of AD requires the presence of both hallmarks, as outlined in our Introduction. Moreover, previous studies (as well as our data) suggest that CD8⁺ T cells have a role in the induction of tau pathology, but this aspect was not considered by Altendorfer and colleagues, thus casting some doubt on the relevance of their results in AD (Laurent, C. *et al.* 2017; Merlini, M *et al.* 2018). In our study, we used the 3xTg-AD mouse model, which develops both amyloid and tau pathologies, mimicking human AD more closely. Finally, Altendorfer and colleagues used old mice (24-27 months of age), in which the distribution and function of brain CD8⁺ T cells may be affected by the ageing process. Accordingly, CD8⁺ T cells accumulate in the old brain, favoring axonal degeneration and motor decline (Groh J. *et al.*, 2021). Instead, we focused on mice at the beginning of cognitive deficit (6 months of age), thus excluding aging-dependent CD8⁺ T cell alterations. We believe that the apparent discrepancies between our study and Altendorfer are due to the type of investigated tissue (brain and meninges processed together versus tissue separation in our study), the mouse model of amyloidosis versus amyloidosis and tau, and the mouse age.

In the Kimura study (Kimura, K. *et al.* 2023), the authors evaluated the percentage of CD69⁺CD103⁺ cells in the CSF of a small group of AD patients calculated on the total memory CD45RA⁻CD8⁺ T cell population and not on the population of Trm cells used in our work. Moreover, Kimura eFigure 3 shows a similar frequency of CD69⁺CD103⁺ cells in the RA⁻ and RA⁺ populations, but the authors show no significant difference in the percentage of CD69⁺CD103⁺ cells in the memory CD45RA⁺CCR7⁻CD8⁺ T cell population in the CSF of AD patients in comparison to control subjects. This strongly suggests that the results may be different if the frequency of CD69⁺CD103⁺ Trm CD8⁺ T cells is calculated on the Trm population, as in our study. Moreover, our data in Fig. 6a-h provide a reanalysis of a CSF dataset (Gate, D. *et al.* 2020) and show the presence of a consistent population of KLF2⁺ circulating cells in the CSF of human subjects (57.5% of the total CD8⁺ population) whereas only 42.5% were Trm cells, further suggesting that the Kimura study

may have yielded different results if focused solely on the Trm population. Notably, the data reported by Smolders *et al.* (2018) consistently show that the majority of intraparenchymal CD8⁺ T cells are Trm cells, thus clearly differing them from the CSF CD8⁺ T cell composition. Thus, we believe that the Kimura study clearly differs from our data in terms of the target site (CSF versus parenchyma) and population (CD45RA⁻ versus classical Trm cells). We discuss these data in the revised manuscript (p16).

9- While GrK is implicated as a neurotoxic mediator, the exact intracellular and downstream signaling pathways triggered by GrK–PAR-1 activation are not detailed. For instance, does GrK lead to oxidative stress, synaptic pruning, or other specific neurotoxic effects?

To understand downstream signaling pathways induced by GrK, we analyzed the proteome of differentiated human neuroblastoma SH-SY5Y cells (treated with recombinant GrK or untreated) in the presence or absence of the PAR-1 inhibitor SCH79797. The results, now shown in Fig. 7h, i and described on p13, showed that GrK significantly increased the abundance of 84 proteins, including proteins involved in the development of AD (Supporting Information 4). Pathway enrichment analysis applied to these proteins revealed that the “Alzheimer’s/Neurodegeneration” and “Kinase pathway” clusters were the most enriched (Fig. 7h). In the “Alzheimer’s/Neurodegeneration” cluster, the most significantly enriched pathways included “amyloid fiber formation”, “neurodegenerative diseases”, and “deregulated CDK5 triggers multiple neurodegeneration” (Fig. 7i, Supporting Information 4). The “Kinase pathway” cluster of terms was characterized by the upregulation of pathways related to the activation of MAP kinases (Mazanetz, M.P *et al.*, 2007), which mediate tau hyperphosphorylation, and the “Post-translational protein phosphorylation pathway” (Fig. 7i). We also found that GrK induced the upregulation of “NF-κB signaling”, which in neurons is associated with the modulation of ion channel protein expression, supporting Ca²⁺ increases at intraneuronal levels, and with a higher Aβ load (Park, K.M. *et al*, 2010). Notably, the induction of all these pathways was prevented when GrK treatment took place in the presence of SCH79797, leading to the significant upregulation of only 12 proteins (Supporting Information 4), whereas enrichment analysis detected no pathways significantly related to these proteins. Overall, our new proteomic data bring substantial novelty and show that GrK promotes AD-related neuropathological alterations. We thank the reviewer for this comment, which helped us to improve our work.

Minor

1- In the introduction line 78, Temra cells should be defined as C45RA+CCR7-

We have refined the description of the cells as requested (p3).